# Multiscale Training of Convolutional Neural Networks

**Shadab Ahamed**  *shadab.ahamed@hotmail.com*
*Department of Physics & Astronomy,*
*University of British Columbia*

**Niloufar Zakariaei**  *nilouzk@student.ubc.ca*
*Department of Earth, Ocean and Atmospheric Sciences,*
*University of British Columbia*

**Eldad Haber**  *ehaber@eoas.ubc.ca*
*Department of Earth, Ocean and Atmospheric Sciences,*
*University of British Columbia*

**Moshe Eliasof**  *me532@cam.ac.uk*
*Department of Applied Mathematics and Theoretical Physics,*
*University of Cambridge*

**Reviewed on OpenReview:** *https://openreview.net/forum?id=HTQuEZwEHw*

## Abstract

Training convolutional neural networks (CNNs) on high-resolution images is often bottle-necked by the cost of evaluating gradients of the loss on the finest spatial mesh. To address this, we propose *Multiscale Gradient Estimation (MGE)*, a Multilevel Monte Carlo-inspired estimator that expresses the expected gradient on the finest mesh as a telescopic sum of gradients computed on progressively coarser meshes. By assigning larger batches to the cheaper coarse levels, MGE achieves the same variance as single-scale stochastic gradient estimation while reducing the number of fine mesh convolutions by a factor of 4 with each downsampling. We further embed MGE within a *Full-Multiscale* training algorithm that solves the learning problem on coarse meshes first and "hot-starts" the next finer level, cutting the required fine mesh iterations by an additional order of magnitude. Extensive experiments on image denoising, deblurring, inpainting and super-resolution tasks using UNet, ResNet and ESPCN backbones confirm the practical benefits: Full-Multiscale reduces the computation costs by 4-16× with no significant loss in performance. Together, MGE and Full-Multiscale offer a principled, architecture-agnostic route to accelerate CNN training on high-resolution data without sacrificing accuracy, and they can be combined with other variance-reduction or learning-rate schedules to further enhance scalability.[1]

## 1 Introduction

In this work, we consider the task of learning a functional $y(\mathbf{x}) = \phi(u(\mathbf{x}))$, where $\mathbf{x}$ is a position (in 2D $\mathbf{x} = (\mathbf{x}_1, \mathbf{x}_2)$, $u(\mathbf{x}) \in \mathcal{U}$ and $y(\mathbf{x}) \in \mathcal{Y}$ are families of functions. To this end, let us assume that we have discrete samples from $\mathcal{U}$ and $\mathcal{Y}$, that is $\mathbf{u}_i^h = u_i(\mathbf{x}_h), \mathbf{y}_i^h = \phi(u_i(\mathbf{x}_h))$ for $i = 1, \ldots, M$, associated with some resolution $h$. A common approach to learning the function is to parameterize the problem, typically by a deep network, and replace $\phi$ with a function $f(\cdot, \cdot)$ that accepts the vector $\mathbf{u}^h$ and learnable parameters $\boldsymbol{\theta}$ which leads to the problem of estimating $\boldsymbol{\theta}$ such that $\mathbf{y}_i^h \approx f(\mathbf{u}_i^h, \boldsymbol{\theta})$ for $i = 1, \ldots, M$. To evaluate $\boldsymbol{\theta}$, the following stochastic optimization problem is formed and solved,

$$\min_{\boldsymbol{\theta}} \ \mathbb{E}_{\mathbf{u}^h, \mathbf{y}^h} \ell\left(f(\mathbf{u}^h, \boldsymbol{\theta}), \mathbf{y}^h\right), \tag{1}$$

---

[1]Code available at: https://github.com/ahxmeds/multiscale-gradient-estimation

where $\ell(\cdot, \cdot)$ is a loss function (typically mean squared error). Standard approaches use variants of stochastic gradient descent (SGD) to estimate the loss and its gradient for different samples of $(\mathbf{u}^h, \mathbf{y}^h)$. In deep learning with convolutional neural networks, the parameter $\boldsymbol{\theta}$ (the convolutional weights) has identical dimensions, independent of the resolution. Although variants of SGD (e.g. Adam, AdamW, etc.) are widely used, their computational cost can become prohibitively high as the mesh-size $h$ decreases, especially when evaluating the function $f$ on a fine mesh for many samples $\mathbf{u}_i^h$. This challenge is worsened if the initial guess $\boldsymbol{\theta}$ is far from optimal, requiring many costly iterations for large data sizes $M$. One way to avoid large meshes is to use small crops of the data where large images are avoided, however, this can degrade performance, especially when a large receptive field is required for learning (Araujo et al., 2019).

**Background and Related Work.** Computational cost reduction can be achieved by leveraging different resolutions, a fundamental concept to multigrid and multiscale methods. These methods have a long history of solving partial differential equations and optimization problems (Trottenberg et al., 2001; Briggs et al., 2000; Nash, 2000). Techniques like multigrid (Trottenberg et al., 2001) and algorithms such as MGopt (Nash, 2000; Borzi, 2005) and Multilevel Monte Carlo (Giles, 2015; 2008; Van Barel & Vandewalle, 2019) are widely used for optimization and differential equations.

In deep learning, multiscale or pyramidal approaches have been used for image processing tasks such as object detection, segmentation, and recognition, where analyzing multiple resolutions is key (Scott & Mjolsness, 2019; Chada et al., 2022; Elizar et al., 2022). Recent methods improve standard CNNs for multiscale computations by introducing specialized architectures and training methods. For instance, the work by He & Xu (2019) uses multigrid methods in CNNs to boost efficiency and capture multiscale features, while Eliasof et al. (2023b) focuses on multiscale channel space learning, and van Betteray et al. (2023) unifies both. Li et al. (2020) introduced the Fourier Neural Operator, enabling mesh-independent computations, and Wavelet neural networks were explored to capture multiscale features via wavelets (Fujieda et al., 2018; Finder et al., 2022; Eliasof et al., 2023a).

While often overlooked, it is important to note that these approaches, can be divided into two families of approaches that leverage multiscale concepts. The *first* is to learn parameters for each scale, and a separate set of parameters that mix scales, as in UNet (Ronneberger et al., 2015). The *second*, called *multiscale training*, enables the approximation of fine-scale parameters using coarse-scale samples (Haber et al., 2017b; Wang et al., 2020; Ding et al., 2020; Ganapathy & Liew, 2008). The second approach aims to gain computational efficiency, as it approximates fine mesh parameters using coarse meshes, and it can be coupled with the first approach, and in particular with UNets.

**Our approach.** This work advances the multiscale training paradigm by rigorously adapting Multilevel Monte Carlo (MLMC) methods to the non-convex landscape of CNN training. While concepts like MLMC and multigrid are well-established in numerical analysis, their application to stochastic gradient estimation for deep CNNs is non-trivial and requires specific theoretical justification. We distinguish our contributions beyond a simple recombination of existing methods in three key ways.

First, we establish the theoretical bounds for CNN gradient estimation. We explicitly derive the error bounds for the Multiscale Gradient Estimation (MGE) estimator, proving that under standard Lipschitz conditions, the difference between gradients on fine and coarse meshes decays as $\mathcal{O}(h)$. This provides the necessary theoretical guarantee for applying variance reduction to convolutional weights, ensuring that the optimization does not diverge when mixing scales, a specificity not addressed in standard MLMC literature.

Second, we provide a rigorous analysis of subsampling strategies. Unlike prior empirical works (Haber et al., 2017a), we mathematically demonstrate that the choice of downsampling is critical: we prove that coarsening-based subsampling yields an error that vanishes as resolution increases ($\mathcal{O}(2^L h)$), whereas cropping-based strategies result in a constant error bound ($\mathcal{O}(1)$) regardless of resolution. This offers a principled guideline for multiscale training paradigm design.

Finally, we embed these theoretical insights into a Full-Multiscale training algorithm. By combining the variance reduction of MGE with a coarse-to-fine "hot-start" initialization, we demonstrate an architecture-agnostic framework that accelerates training by orders of magnitude for tasks ranging from denoising to super-resolution, without significant loss in performance over key metrics.

Although our method exploits the convolutional operation, it does not explicitly address alternative mechanisms like attention. Consequently, while the framework could, in principle, be adapted to attention, it does not offer the same theoretical guarantees that we establish for convolutions.

Our main contributions are: (i) we propose a new multiscale training algorithm, *Multiscale Gradient Estimation (MGE)*, deriving explicit error bounds for gradient convergence in convolutional networks; (ii) we theoretically analyze the limitations of image subsampling strategies, rigorously proving why coarsening outperforms cropping within a multiscale framework; and (iii) we validate our approach via a *Full-Multiscale* training algorithm on benchmark tasks, showcasing enhanced computational efficiency and scalability across diverse architectures.

## 2    Multiscale Gradient Estimation

We now present the standard approach of training CNNs, identify its major computational bottleneck, and propose a novel approximation to the gradient that can be used for different variants of SGD.

**Standard Training of Convolutional Networks.** Suppose that we use a gradient descent-based method to train a CNN with input resolution $h^2$ and with trainable parameters $\boldsymbol{\theta}$. Under gradient-based methods, the parameters $\boldsymbol{\theta}$ can be learned iteratively using,

$$\boldsymbol{\theta}_{k+1} = \boldsymbol{\theta}_k - \mu_k \mathbb{E}_{\mathbf{u}^h, \mathbf{y}^h} \left[ \mathbf{g}(\mathbf{u}^h, \mathbf{y}^h, \boldsymbol{\theta}_k) \right], \tag{2}$$

where $\mathbf{g}(\mathbf{u}^h, \mathbf{y}^h, \boldsymbol{\theta}_k)$ at iteration $k$ represents the gradient with respect to the parameters $\boldsymbol{\theta}$ of some loss function $\ell$ (e.g., the mean squared error function) given by $\mathbf{g}(\mathbf{u}^h, \mathbf{y}^h, \boldsymbol{\theta}_k) = \nabla_{\boldsymbol{\theta}} \ell \left( f(\mathbf{u}^h, \boldsymbol{\theta}_k), \mathbf{y}^h \right)$ and $\mu_k$ represents the learning rate. The expectation $\mathbb{E}$ is taken with respect to the input-label pairs $(\mathbf{u}^h, \mathbf{y}^h)$. Evaluating the expected value of the gradient can be highly expensive, especially on fine meshes where the value of $h$ is very small. To understand why, consider the estimation of the expected value of the gradient using sample mean of $\mathbf{g}$ with a batch of $N$ samples,

$$\mathbb{E}_{\mathbf{u}^h, \mathbf{y}^h} \left[ \mathbf{g}(\mathbf{u}^h, \mathbf{y}^h, \boldsymbol{\theta}_k) \right] \approx \frac{1}{N} \sum_i \mathbf{g}(\mathbf{u}_i^h, \mathbf{y}_i^h, \boldsymbol{\theta}_k). \tag{3}$$

The above approximation results in an error in the gradient. Under some mild assumptions on the sampling of the gradient value (Johansen et al., 2010), the error can be bounded by,

$$\left\| \mathbb{E} \left[ \mathbf{g}(\mathbf{u}^h, \mathbf{y}^h, \boldsymbol{\theta}_k) \right] - \frac{1}{N} \sum_i \mathbf{g}(\mathbf{u}_i^h, \mathbf{y}_i^h, \boldsymbol{\theta}_k) \right\| \leq \frac{C}{\sqrt{N}}, \tag{4}$$

for some constant $C$, where $\| \cdot \|$ represent the $L^2$ norm. Clearly, obtaining an accurate evaluation of the gradient (that is, with low variance) requires sampling $\mathbf{g}(\mathbf{u}_i^h, \mathbf{y}_i^h, \boldsymbol{\theta}_k)$ across many data points $i$ with sufficiently small $h$ (high-resolution). This tradeoff between the sample size $N$ and the accuracy of the gradient estimation, is the costly part of training a deep network on high-resolution data.

To alleviate the problem, it is common to use large batches, effectively enlarging the sample size. It is also possible to use various variance reduction techniques (Anschel et al., 2017; Chen et al., 2017; Alain et al., 2015). Nonetheless, for high-resolution images, or 3D inputs, the large memory requirement limits the size of the batch. A small batch size can result in noisy, highly inaccurate gradients, and slow convergence (Shapiro et al., 2009).

### 2.1    Efficient Training with Multiscale Estimation of the Gradient

To reduce the cost of the computation of the gradients, we use a classical trick proposed in the context of Multilevel Monte Carlo methods (Giles, 2015). To this end, let $h = h_1 < h_2 < ... < h_L$ be a sequence of mesh

---

[2]In this paper, we define resolution $h$ as the pixel size on a 2D uniform mesh grid, where smaller $h$ indicates higher resolution. For simplicity, we assume the same $h$ across all dimensions, though different resolutions can be assigned per dimension.

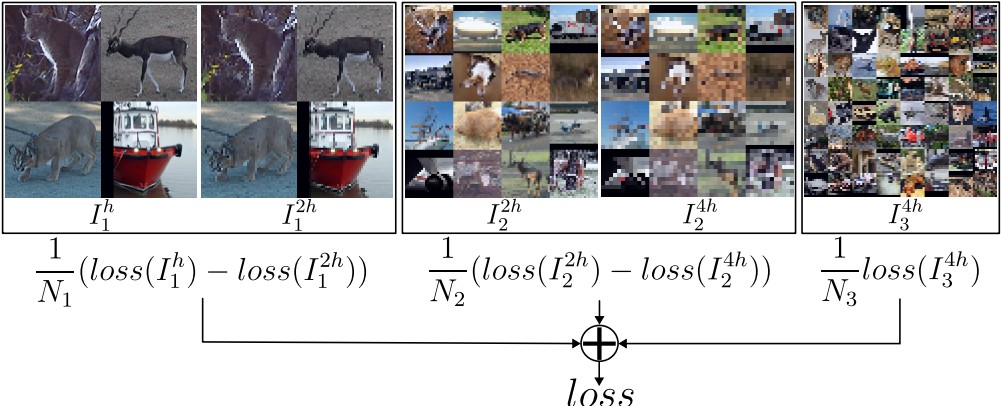

$$\frac{1}{N_1}(loss(I_1^h) - loss(I_1^{2h})) \qquad \frac{1}{N_2}(loss(I_2^{2h}) - loss(I_2^{4h})) \qquad \frac{1}{N_3}loss(I_3^{4h})$$

$$\bigoplus$$

$$loss$$

Figure 1: Illustration of our Multiscale Gradient Estimation (MGE) algorithm introduced in Section 2. This figure shows a schematic of a 3-level MGE algorithm with resolutions $h$ (finest), $2h$, and $4h$ (coarsest) with batch sizes $N_3 > N_2 > N_1$.

sizes, for which the functions $u$ and $y$ are discretized on. We can easily sample (or coarsen) $u$ and $y$ to some mesh $h_j, 1 \leq j \leq L$. We consider the following identity, based on the telescopic sum and the linearity of the expectation,

$$\mathbb{E}\left[\mathbf{g}^{h_1}(\boldsymbol{\theta})\right] = \mathbb{E}\left[\mathbf{g}^{h_L}(\boldsymbol{\theta})\right] + \mathbb{E}\left[\mathbf{g}^{h_{L-1}}(\boldsymbol{\theta}) - \mathbf{g}^{h_L}(\boldsymbol{\theta})\right] + \ldots + \mathbb{E}\left[\mathbf{g}^{h_1}(\boldsymbol{\theta}) - \mathbf{g}^{h_2}(\boldsymbol{\theta})\right], \tag{5}$$

where for shorthand we define the gradient of the loss with respect to $\boldsymbol{\theta}$ with resolution $h_j$ by $\mathbf{g}^{h_j}(\boldsymbol{\theta}) = \mathbf{g}(\mathbf{u}^{h_j}, \mathbf{y}^{h_j}, \boldsymbol{\theta})$. The core idea of our Multiscale Gradient Estimation (MGE) approach, is that *the expected value of each term in the telescopic sum is approximated using a different batch of data with a different batch size.* This concept is demonstrated in Figure 1 and can be written as,

$$\mathbb{E}\left[\mathbf{g}^{h_1}(\boldsymbol{\theta})\right] \approx \frac{1}{N_L}\sum_i \mathbf{g}_i^{h_L}(\boldsymbol{\theta}) + \sum_{j=2}^{L} \frac{1}{N_{j-1}}\sum_i \left(\mathbf{g}_i^{h_{j-1}}(\boldsymbol{\theta}) - \mathbf{g}_i^{h_j}(\boldsymbol{\theta})\right) \tag{6}$$

To understand why this concept is beneficial, we analyze the error obtained by sampling each term in Equation (6). Evaluating the first term in the sum requires evaluating the function $\mathbf{g}$ on the coarsest mesh $h_L$ (i.e., the lowest resolution) using downsampled inputs. Therefore, it can be efficiently computed, while utilizing a large batch size $N_L$. Thus, following Equation (4), the approximation error of the first term in Equation (6) can be bounded by,

$$\left\|\mathbb{E}\left[\mathbf{g}^{h_L}(\boldsymbol{\theta})\right] - \frac{1}{N_L}\sum_i \mathbf{g}_i^{h_L}(\boldsymbol{\theta})\right\| \leq \frac{C}{\sqrt{N_L}}. \tag{7}$$

Throughout our discussion, we assume that the per-pixel gradient contribution $\boldsymbol{\varphi}(u^h)$ is uniformly bounded and Lipschitz-continuous in its argument, so that $\|\boldsymbol{\varphi}(u^h) - \boldsymbol{\varphi}(v^h)\|$ grows at most linearly with $\|u^h - v^h\|$. In addition, when relating gradients across resolutions, we assume that each coarse-grid representative $c_{2h}^k$ (see Appendix A.1 for a detailed discussion) approximates all fine-grid values $u^h$ within its block to within $\mathcal{O}(h)$, reflecting the smoothness of the underlying image signal. Additionally, we need to evaluate the terms of the form,

$$\mathbf{r}_j = \mathbb{E}\left[\mathbf{g}^{h_{j-1}}(\boldsymbol{\theta}) - \mathbf{g}^{h_j}(\boldsymbol{\theta})\right]. \tag{8}$$

The above $\mathbf{r}_j$ can similarly be approximated by sampling with a batch size $N_{j-1}$,

$$\hat{\mathbf{r}}_j = \frac{1}{N_{j-1}}\sum_i \left(\mathbf{g}_i^{h_{j-1}}(\boldsymbol{\theta}) - \mathbf{g}_i^{h_j}(\boldsymbol{\theta})\right), \tag{9}$$

for $j = 2, \ldots, L$. The key question is: what is the error in approximating $\mathbf{r}_j$ by the finite sample estimate $\hat{\mathbf{r}}_j$? Previously, we focused on error due to sample size $N$. However, note that the exact term $\mathbf{r}_j$ is computed by

evaluating $\mathbf{g}$ on two resolutions of the **same samples** and subtracting the results. The key observation is that if the evaluation of $\mathbf{g}$ on different resolutions yields similar results, then $\mathbf{g}$ computed on a mesh with step size $h_j$ can be utilized to approximate the gradient $\mathbf{g}$ on a mesh with finer resolution $h_{j-1}$, making the approximation error $\hat{\mathbf{r}}_j$ small. Furthermore, for the multiscale estimator we adopt the standard cross-resolution discrepancy condition,

$$\|\mathbf{g}_i^{h_{j-1}}(\boldsymbol{\theta}) - \mathbf{g}_i^{h_j}(\boldsymbol{\theta})\| \leq Bh_{j-1}^p, \tag{10}$$

for some constants $B > 0$ and $p > 0$ both independent of the pixel-size $h_{j-1}$. This captures the fact that evaluating the gradient on a slightly coarser mesh yields an approximation whose accuracy improves as the resolution is refined. Then, we can bound the error of approximating $\mathbf{r}_j$ by $\hat{\mathbf{r}}_j$ as follows,

$$\|\mathbf{r}_j - \hat{\mathbf{r}}_j\| \leq BC\frac{h_{j-1}^p}{\sqrt{N_{j-1}}}. \tag{11}$$

Note that, under the assumption that Equation (10) holds, the gradient approximation error between different resolutions decreases as the resolution increases (i.e., $h \to 0$). Combining the terms, the sum of the gradient approximation obtained from the telescopic sum in Equation (6) can be bounded by,

$$e = C \left( \frac{1}{\sqrt{N_L}} + B \sum_{j=2}^{L} \frac{h_{j-1}^p}{\sqrt{N_{j-1}}} \right). \tag{12}$$

*Sketch of derivation for Equation* (12)*:* Starting from the telescoping representation of the fine-scale gradient, we decompose the total error of the multiscale estimator into the sampling error at the coarsest level and the sampling errors of the correction terms across levels. Under our boundedness and Lipschitz assumptions on the per-pixel gradient contribution, the coarsest-level term behaves like a standard Monte Carlo estimator and contributes a term proportional to $1/\sqrt{N_L}$. For each correction term, the cross-resolution discrepancy condition $\|g^{h_{j-1}} - g^{h_j}\| \leq Bh_{j-1}^p$ implies that its sampling error is bounded by a factor $h_{j-1}^p/\sqrt{N_{j-1}}$. Summing these contributions over all levels and absorbing constants yields the bound in Equation (12). In the two-level case as discussed next, we can then specialize this expression, enforce that the multiscale estimator matches the single-scale error level, and choose a simple allocation between coarse and fine batches, which leads directly to the form stated in Equation (13).

**Computational Complexity of using Multiscale Gradient Estimation.** Let us look at an exemplary 2-level case (using 2 mesh grids $h_1$ and $h_2$ with $h = h_1 < h_2$) used for MGE. A standard single-scale gradient estimation (on fine mesh $h_1$) with $N$ samples yields an accuracy of $e_N = C/\sqrt{N}$. If we are to achieve the same accuracy using a 2-level method, then, using Equation (12) and choosing $N_2 = 4N_1$ (that is making the batch size of the coarse mesh $h_2$ four times larger than the one on the fine mesh $h_1$), we see that using MGE the same error is obtain by sampling the fine mesh

$$N_1 = \frac{1}{4}(1 + 2Bh^p)^2 N. \tag{13}$$

For high-resolution images, as the mesh size $h \to 0$ and $B$ is bounded, $Bh^p \ll 1$ and therefore the number of computations on high-resolution samples in a 2-level MGE is approximately $1/4$ compared to a single mesh algorithm. To see the effect of this sampling on the overall cost, similar to other multiscale algorithms (see Trottenberg et al. (2001)), it is useful to define a quantity *workunit* (#WU). A workunit is defined as the cost of computation of the convolution operation on the finest mesh. For single-scale estimation of gradients, an error of $e_N$ is achieved using $N$ #WU. Let us analyze the same for the 2-level MGE algorithm, which involves the computation of the loss over three different batches: a batch of $N_2 = 4N_1 \approx N$ samples over the coarse grid for the first term in Equation (12), a second batch of $N_1 \approx N/4$ samples over the coarse mesh and finally a third batch of $N_1 \approx N/4$ over the fine mesh. Since, the cost of convolution of coarse mesh is approximately $1/4$ of the cost of the fine scale convolution, the cost of estimating the gradients using a 2-level MGE is $N \times (\frac{1}{4}) + \frac{N}{4} \times (1 + \frac{1}{4}) = \frac{9N}{16}$ #WU. Thus, even a simple 2-level algorithm can save approximately $43.8\%$ of computations. Detailed calculations for MGE with more levels is provided in Appendix B. When considering the actual wall time for computing multiscale convolutions, note that the two terms (fine and coarse mesh

---

**Algorithm 1** Multiscale Gradient Estimation

---

Set batch size to $N_L$ and sample, $N_L$ samples of $\mathbf{u}^{h_1}$ and $\mathbf{y}^{h_1}$
Pool $\mathbf{u}^{h_L} = \mathbf{R}_{h_1}^{h_L}\mathbf{u}^{h_1}, \quad \mathbf{y}^{h_L} = \mathbf{R}_{h_1}^{h_L}\mathbf{y}^{h_1}$
Set $loss = \ell(\mathbf{u}^{h_L}, \mathbf{y}^{h_L}, \boldsymbol{\theta})$
**for** $j = 1, ..., L$ **do**
    Set batch size to $N_j$ and sample, $N_j$ samples of $\mathbf{u}^{h_1}$ and $\mathbf{y}^{h_1}$
    Pool $\mathbf{u}^{h_j} = \mathbf{R}_{h_1}^{h_j}\mathbf{u}^{h_1}, \quad \mathbf{y}^{h_j} = \mathbf{R}_{h_1}^{h_j}\mathbf{y}^{h_1}$ and $\mathbf{u}^{h_{j-1}} = \mathbf{R}_{h_1}^{h_{j-1}}\mathbf{u}^{h_1}, \quad \mathbf{y}^{h_{j-1}} = \mathbf{R}_{h_1}^{h_{j-1}}\mathbf{y}^{h_1}$
    Compute the losses $\ell(\mathbf{u}^{h_j}, \mathbf{y}^{h_j}, \boldsymbol{\theta})$ and $\ell(\mathbf{u}^{h_{j-1}}, \mathbf{y}^{h_{j-1}}, \boldsymbol{\theta})$
    $loss \leftarrow loss - \ell(\mathbf{u}^{h_j}, \mathbf{y}^{h_j}, \boldsymbol{\theta}) + \ell(\mathbf{u}^{h_{j-1}}, \mathbf{y}^{h_{j-1}}, \boldsymbol{\theta})$
**end for**
Compute the gradient of the loss.

---

computations) can, in principle, be executed in parallel. Therefore, with appropriate parallelization of the two processes, the wall time for a 2-level MGE algorithm could theoretically be reduced to approximately 50% of the time required for a single-scale algorithm. However, it is important to note that in this work, we do not strictly exploit this parallelization potential; our reported experiments are performed sequentially. We leave the engineering of parallel execution to fully realize these theoretical wall-time gains for future work.

Beyond these savings, MGE is easy to implement. It simply requires computing the loss at different input scales and batches. Since gradients are linear, the gradient of the loss naturally yields MGE. The full algorithm is outlined in Algorithm 1.

## 2.2 Multiscale Analysis of Convolutional Neural Networks

We now analyze under which conditions multiscale gradient computation can be applied for convolutional neural network optimization without compromising on its efficiency.

Specifically, for multiscale gradient computation to be effective, the network output and its gradients with respect to the parameters at one resolution should approximate those at another resolution. Here, we explore how a network trained at one resolution $h$, performs on a different resolution $2h$. Specifically, let $f(\mathbf{u}^h, \boldsymbol{\theta})$ be a network that processes images at resolution $h$. The downsampled version $\mathbf{u}^{2h} = \mathbf{R}_h^{2h}\mathbf{u}^h$ is generated via the interpolation matrix $\mathbf{R}_h^{2h}$. We aim to evaluate $f(\mathbf{u}^{2h}, \boldsymbol{\theta})$ using the coarser image $\mathbf{u}^{2h}$. A simple approach is to reuse the parameters $\boldsymbol{\theta}$ from the fine resolution $h$. In Lemma 1 below, we justify such a usage under some conditions.

**Lemma 1 (Convergence of standard convolution kernels).** *Let* $\mathbf{u}^h, \mathbf{y}^h$ *be continuously differentiable grid functions, and let* $\mathbf{u}^{2h} = \mathbf{R}_h^{2h}\mathbf{u}^h$, *and* $\mathbf{y}^{2h} = \mathbf{R}_h^{2h}\mathbf{y}^h$ *be their interpolation on a mesh with resolution* $2h$. *Let* $\mathbf{g}^h$ *and* $\mathbf{g}^{2h}$ *be the gradients of the function in Equation* (6) *with respect to* $\boldsymbol{\theta}$. *Then the difference between* $\mathbf{g}^h$ *and* $\mathbf{g}^{2h}$ *is*

$$\|\mathbf{g}^h - \mathbf{g}^{2h}\| = \mathcal{O}(h).$$

As can be seen in the proof – that we present in Appendix A.1 – of the above lemma, the convergence of the gradient depends on the amount of high frequencies in the data. This makes sense since the restriction $\mathbf{R}_h^{2h}$ damps high frequencies. If the signals $\mathbf{u}^h$ and $\mathbf{y}^h$ contain mainly high frequencies, then the gradients of loss on the coarse mesh cannot faithfully represent those on the fine meshes. We now test the validity of this assumptions for a number of commonly used data set in Example 1.

**Example 1.** *Assume that* $f$ *is a 1D convolution, that is* $f(\mathbf{u}^h, \boldsymbol{\theta}) = \mathbf{u}^h \star \boldsymbol{\theta}$, *and that the loss is a linear model,*

$$\ell_h(\boldsymbol{\theta}) = \frac{1}{n}(\mathbf{u}^h \star \boldsymbol{\theta})^\top \mathbf{y}^h, \tag{14}$$

*where* $\mathbf{y}^h$ *is the discretization of some function* $y$ *on the fine mesh* $h$. *We use a 1D convolution with kernel size* $3 \times 1$, *whose trainable weight vector is* $\boldsymbol{\theta} \in \mathbb{R}^3$ *and compute the gradient of the function on different*

meshes. The loss function on the $i^{th}$ coarse mesh of size $2^i h$ can be written as,

$$\ell_{2^i h}(\boldsymbol{\theta}) = \frac{1}{n_i}(\mathbf{R}_h^{2^i h}\mathbf{u}^h \star \boldsymbol{\theta})^\top(\mathbf{R}_h^{2^i h}\mathbf{y}^h), \tag{15}$$

where $\mathbf{R}_h^{2^i h}$ is a linear interpolation operator that takes the signal from mesh size $h$ to mesh size $2^i h$. We compute the difference between the gradient of the loss function on each level and compute its norm,

$$\delta g = \left\|\boldsymbol{\nabla}_{\boldsymbol{\theta}}\ell_{2^i h} - \boldsymbol{\nabla}_{\boldsymbol{\theta}}\ell_{2^{i+1} h}\right\|. \tag{16}$$

In our experiments, we add Gaussian noise $\mathcal{N}(\mathbf{0}, \mathbf{I})$ scaled by different factors $\sigma$ to the inputs. We compute the values of $\delta g$, and report it in Figure 2. The experiment demonstrates that even for relatively large amounts of noise the difference between the gradients stays rather small. This justified using the approach for a wide range of problems.

Figure 2 also serves as an empirical evaluation of the resolution-induced gradient discrepancy $\|g^{h_{j-1}} - g^{h_j}\|$, which is the key approximation term used in the analysis of Section 2.1. The observed decay of this discrepancy with mesh refinement, as well as its stability under noise perturbations, provides direct support for the cross-resolution consistency assumption underlying the error bounds in Equations (12) and (13). While this does not measure the full gradient error $\|\hat{g} - \nabla f\|$, it verifies precisely the gradient behavior required for the multiscale estimator to be theoretically well-founded.

The analysis suggests that for smooth signals, standard CNNs can be integrated into Multilevel Monte Carlo methods, which form the basis of our MGE. The example demonstrates that while for noisy signals, the difference between gradients on different mesh sizes may not decrease, it is still small and thus can be used for estimating the gradients. We also compute the total wall time (in seconds) for the computation of loss under both the single-scale and MGE frameworks for 512 images from the STL10 dataset in Table 1, where we show that MGE takes considerably lower amount of time as compared to the single-scale operations. All computations were performed on a CPU with 48 cores, and a total available RAM of 819 GB. Note that although in terms of FLOP counts, the computation on a coarser mesh is 4 times more effective, using standard hardware and software (PyTorch) yields more modest gains. This however, can be resolved by designing better hardware and software implementations.

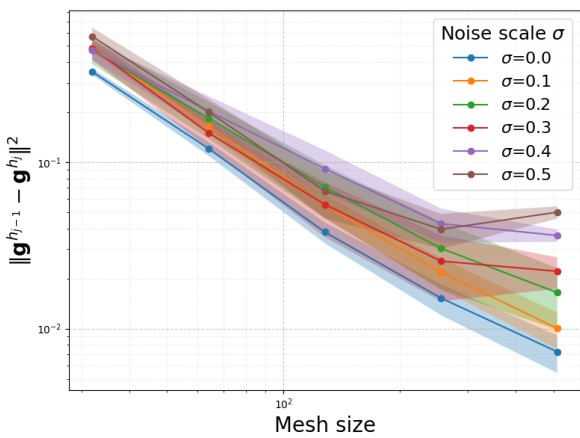

Figure 2: The difference between the gradients on different mesh sizes and for different noise levels $\sigma$. The difference remains small even for large noise levels.

Table 1: Comparing the time for loss evaluation on images of different sizes for single images, number of loss function evaluations and total evaluation time for loss computation under single-scale and MGE over 512 images from the STL10 dataset. For this example, all evaluations were performed using a UNet. For the telescopic sum of MGE, for each $n$, the batch size at the finest possible resolution was set to 16 and subsequent coarser levels were assigned batch sizes in even multiples of 16 (32, 64, etc.) and the coarsest level got all the remaining (out of 512) number of images.

| Mesh size $(n^2) \rightarrow$ | | $256^2$ | $128^2$ | $64^2$ | $32^2$ | $16^2$ |
|---|---|---|---|---|---|---|
| Wall time for loss computation on a single image (s) | | 3.338 | 1.502 | 0.956 | 0.662 | 0.458 |
| Number of loss evaluations | Single-scale | 512 | 512 | 512 | 512 | 512 |
| | MGE | 752 | 624 | 560 | 528 | 512 |
| Wall time on 512 images (s) | Single-scale | 1709.06 | 769.02 | 489.47 | 338.94 | 234.50 |
| | MGE | 527.58 | 345.98 | 274.24 | 245.09 | 234.50 |

## 3  The Full-Multiscale Training Algorithm

While MGE can accelerate the computation of the gradient, solving the optimization problem is still computationally expensive. MGE makes each iteration computationally cheaper, nonetheless, the number of

iterations needed for the solution of the optimization problem is typically very high. Many problems require thousands, if not tens of thousands of iterations, where each iteration sees only a small portion of the data (a batch), due to the large size of the whole data set. Multiscale framework can be leveraged to dramatically reduce the cost of the optimization problem. A common method is to first coarsen the images in the data and solve the optimization problem (that is, estimate the parameters, $\boldsymbol{\theta}$) where the data is interpolated (pooled) to a coarser mesh. Since the loss and gradients of loss on the coarse mesh approximates the fine mesh problem, the output of the optimization problem on coarse mesh serves as a very good initial guess to the solution of the fine scale problem. When starting the solution close to its optimal value, one requires only few iterations on the fine mesh to converge. This process is often referred to as called *mesh homotopy* (Haber et al., 2007). This resolution-dependent approach to training CNNs is summarized in Full-Multiscale in Algorithm 2 (Borzì & Schulz, 2012).

---

**Algorithm 2** Full-Multiscale

Randomly initialize the trainable parameters $\boldsymbol{\theta}_*^H$.
**for** $j = 1, ..., L$ **do**
    Set mesh size to $h_j = 2^{L-j}h$ and $\boldsymbol{\theta}_0^{h_j} = \boldsymbol{\theta}_*^H$.
    Solve the optimization problem on mesh $h_j$ for $\boldsymbol{\theta}_*^{h_j}$.
    Set $\boldsymbol{\theta}_*^H \leftarrow \boldsymbol{\theta}_*^{h_j}$.
**end for**

---

**Convergence rate for Full-Multiscale.** To further understand the effect of the Full-Multiscale approach, we recall the stochastic gradient descent (SGD) converge rate. For the case where the learning rate converges to 0 and $\ell$ is smooth and convex, we have that after $k$ iterations of SGD, we can bound the error of the loss by,

$$\left\| \mathbb{E}\left[\ell(\boldsymbol{\theta}_k)\right] - \mathbb{E}\left[\ell(\boldsymbol{\theta}_*^h)\right] \right\| \leq C_0 \frac{C}{k}, \tag{17}$$

where $C$ is a constant, $\boldsymbol{\theta}_*^h$ is the parameter that optimizes the expectation of the loss, and $C_0$ is a constant that depends on the initial error at $\boldsymbol{\theta}_0$. Let $\boldsymbol{\theta}_*^H$ and $\boldsymbol{\theta}_*^h$ be the parameters that minimize the expectation of the loss for meshes with resolution $H$ and $h$, respectively with $H > h$ and assume that $\|\boldsymbol{\theta}_*^H - \boldsymbol{\theta}_*^h\| \leq \gamma H$,

where $\gamma$ is a constant independent of $h$. This assumption is standard (see Nash (2000) and it is justified if the loss converges to a finite value as $h \to 0$. During the Full-Multiscale iterations (Algorithm 2), we solve the problem on the coarse mesh $H$ to initialize the fine mesh $h$ solution. Thus, after $k$ steps of SGD on the fine mesh, we can bound the error by,

$$\left\| \mathbb{E}\left[\ell(\boldsymbol{\theta}_k)\right] - \mathbb{E}\left[\ell(\boldsymbol{\theta}_*^h)\right] \right\| \leq \gamma H \frac{C}{k}. \tag{18}$$

Requiring that the error is smaller than some $\epsilon$, renders a bounded number of required iterations

$$k \approx \gamma C \frac{H}{\epsilon}. \tag{19}$$

The above discussion can be summarized by the following theorem.

**Theorem 1 (Hotstarting SGD).** *Let* $f_h(\boldsymbol{\theta}) = \mathbb{E}_{\mathbf{u}^h, \mathbf{y}^h}[\ell(\mathbf{u}^h, \mathbf{y}^h, \boldsymbol{\theta})]$ *and let* $f_H(\boldsymbol{\theta}) = \mathbb{E}_{\mathbf{u}^H, \mathbf{y}^H}[\ell(\mathbf{u}^H, \mathbf{y}^H, \boldsymbol{\theta})]$. *Let* $\boldsymbol{\theta}_*^H = \arg\min_{\boldsymbol{\theta}} f_H(\boldsymbol{\theta})$ *and* $\boldsymbol{\theta}_*^h = \arg\min_{\boldsymbol{\theta}} f_h(\boldsymbol{\theta})$. *Then, the number of iterations needed to optimize* $f_h$ *to an error of* $\epsilon$ *is,*

$$k = \mathcal{O}\left(\frac{H}{\epsilon}\right)$$

In practice, since in Algorithm 2, $H = 2^j h$, the number of iterations for a fixed error $\epsilon$ is halved at each level, the iterations on the finest mesh are a fraction of those on the coarsest mesh which can speed up training by an order of magnitude compared with standard single-scale training.

## 4 Experimental Results and Discussion

In this section, we evaluate the empirical performance of our training strategies: *Multiscale* (Algorithm 1) and *Full-Multiscale* (Algorithm 2), compared to standard Single-scale CNN training.

**Experiments.** We demonstrate the broad application of our Multiscale and Full-Multiscale training strategies using architectures such as ResNet (He et al., 2016), UNet (Ronneberger et al., 2015), and ESPCN (Shi et al., 2016) on a wide variety of tasks ranging from image denoising, deblurring, inpainting, and super-resolution. Additional details on experimental settings, hyperparameters, architectures, and datasets are provided in Appendix D. Notably, recent architectures employ attention. The application of multiscale techniques to attention are beyond the scope of this paper and will be investigated in future work. We also experiment with different approaches to image subsampling that are based either on cropping, coarsening (pooling) or a combination of both within a multiscale training framework based on MGE.

**Research questions.** We seek to address the following questions: (i) How effective are standard convolutions with multiscale training and what are their limitations? (ii) Can multiscale training be broadly applied to typical CNN tasks, and how much computational savings does it offer compared to standard training? (iii) What is the right image subsampling strategy to use, among coarsening and cropping, within a multiscale training framework?

**Metrics.** To address our questions, we focus on performance metrics (e.g., MSE or SSIM) and the computational effort for training as measured by #WU for each method. As a baseline, we train the problem on a single-scale (finest) resolution. As explained in Appendix B, to be unbiased to implementation and software limitations, we define a *workunit* (#WU) as one application of the model on a single image at the finest mesh (i.e., original resolution). In multiscale training, this unit decreases by a factor of 4 with each downsampling by a factor of 2. We then compare #WU across training strategies. We also compare a workunit with computational time. For optimal implementation, there is a linear relationship between the two. Finally, we perform paired t-tests between the Single-scale baseline and each of the Multiscale and Full-Multiscale variants on the test set, using a 5% significance level to assess whether the differences in their mean performance metrics are statistically significant.

Table 2: Comparison of different training strategies, Single-scale, Multiscale and Full-Multiscale (under various image subsampling strategies like coarsening or cropping for the multiscale training), over different networks and across various tasks such as denoising, deblurring, inpainting, and super-resolution. The training computational costs are measured via #WU and the mean performance of the networks on the test set via MSE or SSIM over various tasks. A paired t-test was performed for Multiscale and Full-Multiscale (for coarsening only) with respect to the Single-scale baseline. Here, "*" indicates performance not statistically different from Single-scale, while "**" indicates performance statistically different from Single-scale at a significance level of 5%.

| Training strategy | Subsampling strategy | | #WU ($\downarrow$) | Image Denoising, MSE ($\downarrow$) | |
| | Coarsen | Crops | | UNet | ResNet |
| --- | --- | --- | --- | --- | --- |
| Single-scale | - | - | 480k | 0.1472 | 0.0927 |
| Multiscale (Ours) | ✓ | ✗ | 74k | 0.1417* | 0.0943* |
| Full-Multiscale (Ours) | ✓ | ✗ | 28.7k | 0.1086** | 0.0839** |
| Full-Multiscale (Ours) | ✗ | ✓ | 28.7k | 0.2723 | 0.1599 |
| Full-Multiscale (Ours) | ✓ | ✓ | 28.7k | 0.2632 | 0.1366 |
| | | | | Image Deblurring, MSE ($\downarrow$) | |
| | | | | UNet | ResNet |
| Single-scale | - | - | 480k | 0.1478 | 0.1117 |
| Multiscale (Ours) | ✓ | ✗ | 74k | 0.1420** | 0.1156* |
| Full-Multiscale (Ours) | ✓ | ✗ | 28.7k | 0.1515** | 0.1480** |
| Full-Multiscale (Ours) | ✗ | ✓ | 28.7k | 0.4010 | 0.1852 |
| Full-Multiscale (Ours) | ✓ | ✓ | 28.7k | 0.3541 | 0.1849 |
| | | | | Image Inpainting, SSIM ($\uparrow$) | |
| | | | | UNet | ResNet |
| Single-scale | - | - | 480k | 0.9020 | 0.9079 |
| Multiscale (Ours) | ✓ | ✗ | 74k | 0.8927** | 0.8920** |
| Full-Multiscale (Ours) | ✓ | ✗ | 126k | 0.9112** | 0.9060* |
| Full-Multiscale (Ours) | ✗ | ✓ | 126k | 0.6383 | 0.8527 |
| Full-Multiscale (Ours) | ✓ | ✓ | 126k | 0.6392 | 0.8464 |
| | | | | Image Super-resolution, SSIM ($\uparrow$) | |
| | | | | ESPCN | ResNet |
| Single-scale | - | - | 30k | 0.8147 | 0.8294 |
| Multiscale (Ours) | ✓ | ✗ | 4.6k | 0.7952** | 0.7982** |
| Full-Multiscale (Ours) | ✓ | ✗ | 7.9k | 0.7945** | 0.7929** |
| Full-Multiscale (Ours) | ✗ | ✓ | 7.9k | 0.7590 | 0.7504 |
| Full-Multiscale (Ours) | ✓ | ✓ | 7.9k | 0.7663 | 0.7617 |

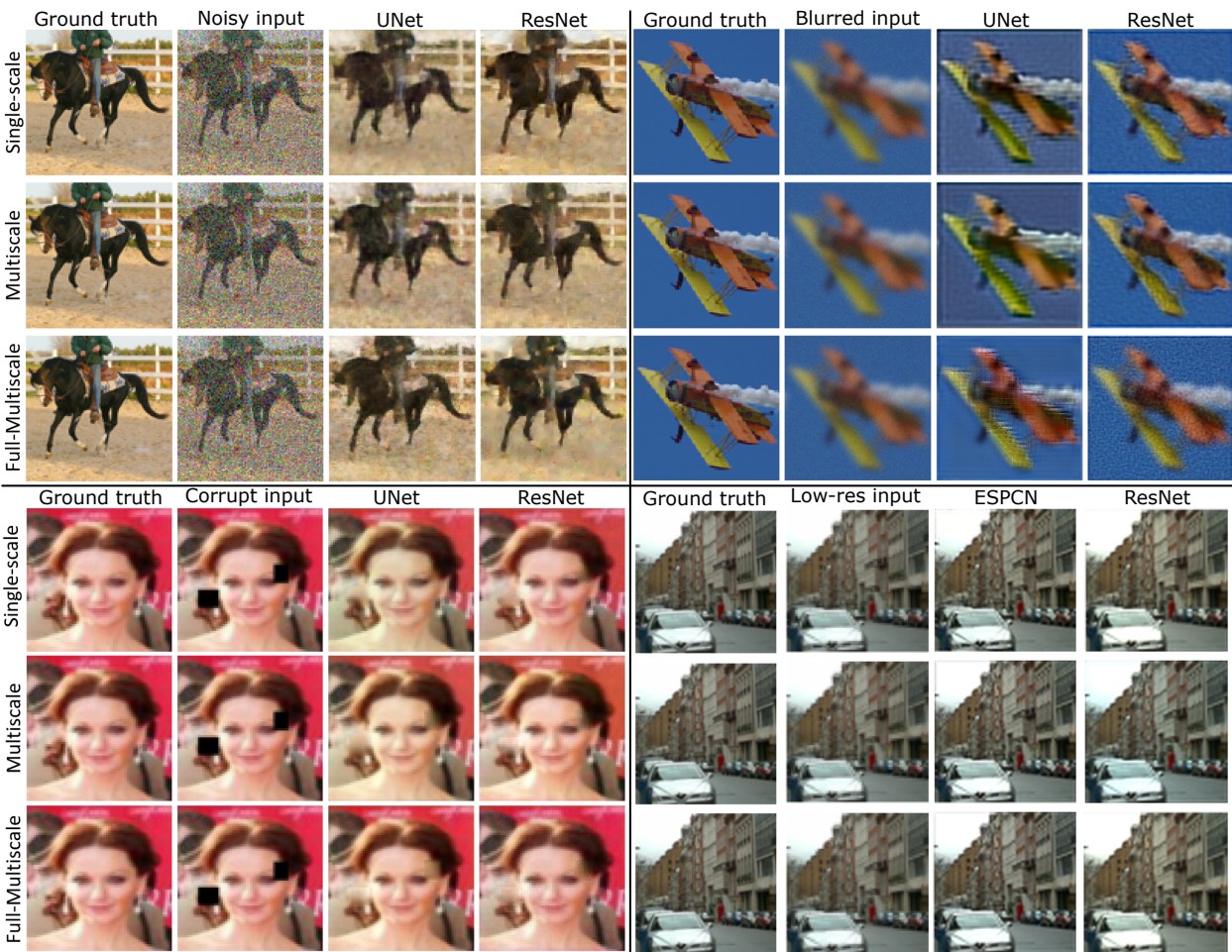

Figure 3: Examples of image recovery over different tasks such as image denoising (top-left), deblurring (top-right), inpainting (bottom-left), and super-resolution (bottom-right) under different training strategies (Single-scale, Multiscale, and Full-Multiscale) using various base networks such as UNet, ResNet, and ESPCN.

**Image denoising.** Here, one assumes data of the form $\mathbf{u}^h = t\mathbf{y}^h + (1-t)\mathbf{z}$, where $\mathbf{y}^h$ is some image on a fine mesh $h$ and $\mathbf{z} \sim \mathcal{N}(0, \mathbf{I})$ is the noise. The noise level $t \in [0, 1]$ is chosen randomly. The goal is to recover $\mathbf{y}^h$ from $\mathbf{u}^h$. The loss to be minimized is $loss(\boldsymbol{\theta}) = \frac{1}{2}\mathbb{E}_{\mathbf{y}^h, \mathbf{u}^h, t}\|f(\mathbf{u}^h, t, \boldsymbol{\theta}) - \mathbf{y}^h\|^2$. In Table 2, training via Multiscale and Full-Multiscale strategies significantly improve training compute (measured by #WU ) while maintaining the quality of the reconstructed denoised image on the STL10 test dataset, measured by MSE. Additional experiments on the CelebA dataset for this task are provided in Appendix C.1 (see, Table 3).

**Image deblurring.** Here, one assumes data of the form $\mathbf{u}^h = \mathbf{K}^h\mathbf{y}^h + \mathbf{z}$, where $\mathbf{y}^h$ is some image on mesh $h$, $\mathbf{K}^h$ is the blurring kernel and $\mathbf{z}$ is the noise. The goal is to recover $\mathbf{y}^h$ from blurred $\mathbf{u}^h$. The loss to be minimized is $loss(\boldsymbol{\theta}) = \frac{1}{2}\mathbb{E}_{\mathbf{y}^h, \mathbf{u}^h}\|f(\mathbf{u}^h, \boldsymbol{\theta}) - \mathbf{y}^h\|^2$. In Table 2, our Full-Multiscale training strategy significantly accelerates training while maintaining the quality of the reconstructed deblurred image on the STL10 dataset, measured by MSE.

**Image inpainting.** Here, one assumes data of the form $\mathbf{u}^h = \mathbf{M}^h\mathbf{y}^h + \mathbf{z}$, where $\mathbf{y}^h$ is a complete image on mesh $h$, $\mathbf{M}^h$ is the image corruption operation and $\mathbf{z}$ is the noise. The goal is to recover $\mathbf{y}^h$ from incomplete noised data $\mathbf{u}^h$. The loss to be minimized is $loss(\boldsymbol{\theta}) = \frac{1}{2}\mathbb{E}_{\mathbf{y}^h, \mathbf{u}^h}\|f(\mathbf{u}^h, \boldsymbol{\theta}) - \mathbf{y}^h\|^2$. In Table 2, our Full-Multiscale training strategy significantly accelerates training while maintaining the quality of the reconstructed inpainted image on the CelebA dataset, measured by SSIM.

**Image super-resolution.** Here, we aim to predict a high-resolution image $\mathbf{u}^h$ from a low-resolution image $\mathbf{y}^l$, which is a downsampled version of $\mathbf{u}^h$. The downsampling process is modeled as $\mathbf{y}^l = \mathbf{D}\mathbf{u}^h + \mathbf{z}$, where $\mathbf{D}(\cdot)$ is a downsampling operator (e.g., bicubic), and $\mathbf{z}$ is noise. The goal is to reconstruct $\mathbf{u}^h$ using a neural network $f(\mathbf{y}^l, \boldsymbol{\theta})$. The loss function is $\mathcal{L}(\boldsymbol{\theta}) = -\mathbb{E}_{\mathbf{y}^l, \mathbf{u}^h} \left[ \text{SSIM}\left(f(\mathbf{y}^l, \boldsymbol{\theta}), \mathbf{u}^h\right) \right]$. In Table 2, our Full-Multiscale training strategy significantly accelerates training while maintaining image quality on the Urban100 dataset, measured by SSIM.

In all our experiments, we noted that within a multiscale training framework, a coarsening-based subsampling strategy is better than a cropping-based or a combination of both coarsening- and cropping-based strategy. We provide a theoretical justification for this observation in Appendix A.2. The error in the estimation of gradient using the telescopic sum in Equation (5) for a coarsening-based subsampling approach varies as $\mathcal{O}(2^L h)$, hence as the mesh resolution $h \to 0$, the error vanishes. On the other hand, for a cropping-based subsampling approach, the error in gradient has an upper bound $2(1 - \frac{m}{N_h})M(L-1)$, where $m$ is the number of pixels in the cropped patch, $N_h$ is the total number of pixels in the entire image, and $M$ represents the upper bound on the norm of gradients for all pixels. Hence, the error in gradients for the cropping-based approach varies as $\mathcal{O}(1)$ (independent of $h$) but grows with the number of levels of resolution $L$ in MGE.

Qualitative performance of these training strategies on different tasks have been presented in Figure 3. We discuss the broader impacts of our work in Section 5. Detailed experimental settings and visualizations for each of the above tasks are provided in Appendices D and E, respectively. Furthermore, we provide additional experiments with deeper networks, comparison under fixed computational budget, and sensitivity to different number of resolution levels for the Multiscale and Full-Multiscale training in Appendices C.2 to C.4, respectively.

**Practical guidelines for hyperparameters ($L$ and batch-size) selection.** While the theoretical derivation of MGE allows for an arbitrary number of levels, practical implementation requires selecting the number of levels $L$ and batch size multipliers (refer to the calculations in Appendix B) based on input resolution and memory constraints. Based on our ablation studies (see Appendix C.4) and variance analysis in , we propose the following guidelines for practitioners:

- *Selecting the number of multiscale levels (L)*: The choice of $L$ involves a trade-off between computational acceleration and approximation error. As the mesh becomes coarser, the spatial structure may degrade to a point where gradients are no longer representative of the fine mesh. We recommend choosing $L$ such that the spatial resolution of the coarsest level $h_L$ remains large enough to capture dominant structural features (typically $\geq 8 \times 8$ pixels). For example, with $64 \times 64$ inputs, we utilized $L = 4$ levels (coarsest resolution $8 \times 8$), but for larger inputs (e.g., $256 \times 256$), $L$ can be increased to 5 or 6 to maximize speedup.

- *Scaling batch sizes*: To maintain the variance reduction properties of the telescopic sum (Equation (13)), batch sizes at coarser levels should ideally increase to compensate for the approximation noise. Since the computational cost of 2D convolutions decreases by a factor of 4 with each downsampling, we recommend increasing the batch size by a factor of 2 to 4 at each subsequent coarser level. This strategy - assigning the largest batches to the cheapest (coarsest) levels - minimizes the variance of the gradient estimator without significantly increasing the wall-clock time or memory footprint.

**Potential challenges with extension to attention-based networks.** While this work demonstrates the efficacy of MGE for convolutional architectures, extending the framework to attention mechanisms presents distinct theoretical challenges primarily due to the global nature of standard self-attention (Vaswani et al., 2017). Our convergence proofs in Lemma 1) rely on the locality of the operator, assuming that gradients on a coarse mesh approximate those on a fine mesh within a bounded local neighborhood. Global attention allows every token to interact with every other token, potentially destabilizing the gradient approximation across scales without specific regularization. Future work addressing this would likely benefit from investigating localized window-based attention mechanisms such as Swin Transformers (Liu et al., 2021), which restore the locality assumption required by our theoretical bounds. Furthermore, successfully adapting MGE to attention offers arguably greater potential rewards: since self-attention computational cost scales quadratically with

resolution, the $4\times$ pixel reduction at each coarser level 2 could theoretically yield up to $16\times$ computational savings per attention layer, significantly amplifying the efficiency gains observed here for convolutions.

## 5 Conclusions

In this paper, we introduced a novel approach to multiscale training for convolutional neural networks, addressing the limitations of high computational costs related to training on single-scale high-resolution data. We theoretically derived error bounds on the expected value of gradients of loss within a multiscale training framework, proved results on the convergence of standard convolutional kernels, discussed the convergence rates of Full-Multiscale algorithm, and provided a theoretical justification for why coarsening is a better image subsampling strategy than cropping within a multiscale training framework. Empirical findings on a number of canonical imaging tasks suggest that our proposed methods with coarsening-based image subsampling can achieve similar or sometimes even better performance than single-scale training with only a fraction of the total training computational costs, as measured by #WUand wall time. While our results indicate that multiscale training has merit using the existing computational framework, we observe a gap between the theoretical complexity to the observed performance. Further gains can be made by using more advanced computational architectures, both in terms of hardware and software. We hope that the theoretical merits discussed in this paper will inspire the development of such efficient implementations that could better utilize multiscale training strategies.

### Broader Impact Statement

Our proposed Multiscale Gradient Estimation and Full-Multiscale training algorithms reduce the number of expensive fine-resolution convolutions by up to $16\times$, which can potentially significantly cut the energy consumption and carbon footprint associated with training high-resolution convolutional networks while preserving accuracy. By lowering the high expense of large-scale training - both in terms of compute hours and electricity consumption - this work has the potential to help make high-fidelity deep learning research more attainable for institutions and researchers facing tight budgetary or infrastructure constraints. At the same time, faster and cheaper training could accelerate the development of powerful models for applications ranging from medical-image reconstruction and diagnosis to environmental sensing and weather forecasting, and also lower the barrier for misuse in areas like pervasive surveillance or deep-fake generation.

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

## A    Proofs

### A.1    Proof to Lemma 1

**Convergence of standard convolution kernels.**    *Let $\mathbf{u}^h, \mathbf{y}^h$ be continuously differentiable grid functions, and let $\mathbf{u}^{2h} = \mathbf{R}_h^{2h} \mathbf{u}^h$, and $\mathbf{y}^{2h} = \mathbf{R}_h^{2h} \mathbf{y}^h$ be their interpolation on a mesh with resolution $2h$. Let $\mathbf{g}^h$ and $\mathbf{g}^{2h}$ be the gradients of the function in Equation (6) with respect to $\boldsymbol{\theta}$. Then the difference between $\mathbf{g}^h$ and $\mathbf{g}^{2h}$ is*

$$\|\mathbf{g}^{2h} - \mathbf{g}^h\| = \mathcal{O}(h).$$

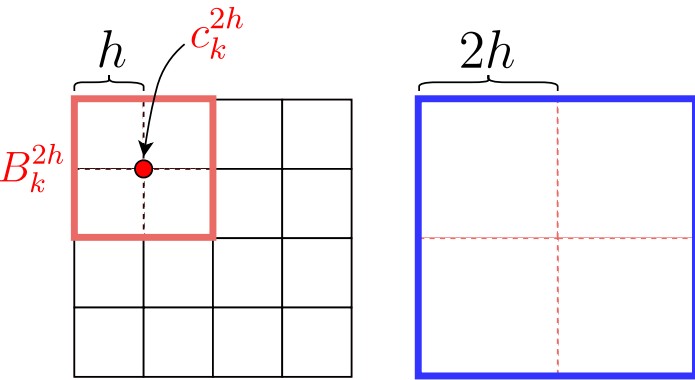

Figure 4: Multiscale Gradient Estimation using coarsening-based subsampling of images. Here, the left figure shows the discretization of an image on a mesh of resolution $h$ (finest mesh). With subsequent coarsening, the image can be downsampled to a mesh of resolution $2h$ (and $4h, 8h$, and so on). For the sake of proving Lemma 1, we define non-overlapping patches $B_k^{2h}$ on the image with resolution $2h$ (with centers $c_k^{2h}$) each of which contains four pixels from the image of resolution $h$. After coarsening, the image looks like the figure on the right with each pixel of resolution $2h$.

*Proof.* To prove Lemma 1, let us assume a mesh with a resolution $h$, as shown in Figure 4 (left) containing $N_h$ pixels. Let the set of all pixels at resolution $h$ be denoted by $\Omega_h$, and $|\Omega_h| = N_h$. While this proof can be generalized for transitions between any two consecutive mesh resolutions $2^{j-1}h \to 2^j h$, here we assume $j = 1$ for brevity and restrict our attention to the coarsening $h \to 2h$ (which is also directly related to the statement of Lemma 1). The extension to arbitrary $j$ follows by the same reasoning.

Let us assume the gradient with respect to the network parameters $\boldsymbol{\theta}$ of the loss function $\ell$ of the form,

$$\boldsymbol{\varphi}(\mathbf{u}_j^h) = \frac{\partial}{\partial \boldsymbol{\theta}} \ell_j(f(\mathbf{u}^h, \boldsymbol{\theta}), \mathbf{y}^h), \tag{20}$$

where $\ell_j(f(\mathbf{u}^h, \boldsymbol{\theta}), \mathbf{y}^h)$ represents the contribution to the loss from the pixel $j$ of the image. To prove this lemma, we make the following three assumptions:

(1) **The gradient $\boldsymbol{\varphi}$ is bounded:** $\|\boldsymbol{\varphi}(x)\| \leq M$, $\forall x \in \Omega_h$

(2) **The gradient $\boldsymbol{\varphi}$ is Lipschitz-continuous:** $\|\boldsymbol{\varphi}(x) - \boldsymbol{\varphi}(y)\| \leq C\|x - y\|$, $\forall x, y \in \Omega_h$, where $C > 0$ is the Lipschitz constant.

(3) **The error in approximating a pixel $x^h$ with $c_k^{2h}$:** As shown in Figure 4 (left), the image on the grid with resolution $h$ can be coarsened to a resolution $2h$. At resolution $2h$, the overall image consists of $N_h/4$ non-overlapping patches $B_k^{2h}$ with their centers denoted as $c_k^{2h}$ for $k \in \{1, \ldots, N_h/4\}$. For a specific $B_k^{2h}$, we assume that the error $\|x^h - c_k^{2h}\| < h$ for $x^h \in B_k^{2h}$ and $k \in \{1, \ldots, N_h/4\}$.

The gradient $\mathbf{g}^h$ can be written as an average over $\boldsymbol{\varphi}(\mathbf{u}_j^h)$ for all $\mathbf{u}_j^h \in \Omega_h$,

$$\mathbf{g}^h = \frac{1}{N_h} \sum_{j=1}^{N_h} \boldsymbol{\varphi}(\mathbf{u}_j^h) = \frac{1}{N_h} \sum_{k=1}^{N_h/4} \sum_{u^h \in B_k^{2h}} \boldsymbol{\varphi}(u^h). \tag{21}$$

Upon coarsening the image to $2h$, the gradient $\mathbf{g}_{\text{coarsen}}^{2h}$ can be written using the midpoints $c_k$ of patches $B_k^{2h}$ as,

$$\mathbf{g}_{\text{coarsen}}^{2h} = \frac{4}{N_h} \sum_{k=1}^{N_h/4} \boldsymbol{\varphi}(c_k). \tag{22}$$

Hence, the residual (Equation (9)) becomes,

$$r_{\text{coarsen}} = \|\mathbf{g}^h - \mathbf{g}_{\text{coarsen}}^{2h}\| = \left\| \frac{1}{N_h} \sum_{k=1}^{N_h/4} \left( \sum_{u^h \in B_k^{2h}} \boldsymbol{\varphi}(u^h) - 4\boldsymbol{\varphi}(c_k) \right) \right\|. \tag{23}$$

Now, using the triangle inequality, we can write,

$$r_{\text{coarsen}} \leq \frac{1}{N_h} \sum_{k=1}^{N_h/4} \left\| \sum_{u^h \in B_k^{2h}} \boldsymbol{\varphi}(u^h) - 4\boldsymbol{\varphi}(c_k) \right\|. \tag{24}$$

But, within a specific $B_k^{2h}$, $\left\| \sum_{u^h \in B_k^{2h}} \boldsymbol{\varphi}(u^h) - 4\boldsymbol{\varphi}(c_k) \right\| = \left\| \sum_{u^h \in B_k^{2h}} \left( \boldsymbol{\varphi}(u^h) - \boldsymbol{\varphi}(c_k) \right) \right\|$. Hence, using the triangle inequality argument again, we have,

$$\left\| \sum_{u^h \in B_k^{2h}} \left( \boldsymbol{\varphi}(u^h) - \boldsymbol{\varphi}(c_k) \right) \right\| \leq \sum_{u^h \in B_k^{2h}} \left\| \boldsymbol{\varphi}(u^h) - \boldsymbol{\varphi}(c_k) \right\| \leq \sum_{u^h \in B_k^{2h}} C\|u^h - c_k\| \leq 4Ch, \tag{25}$$

where in the last two steps in the above expression, we have utilized assumptions (2) and (3). Finally, we have,

$$r_{\text{coarsen}} \leq \frac{1}{N_h} \sum_{k=1}^{N_h/4} 4Ch = Ch. \tag{26}$$

This implies that,

$$\boxed{r_{\text{coarsen}} = \|\mathbf{g}^h - \mathbf{g}_{\text{coarsen}}^{2h}\| = \mathcal{O}(h)} \tag{27}$$

which completes the proof. Although our proofs use a single convolution for simplicity and clarity, the key bound $\|g^{h_{j-1}} - g^{h_j}\| = O(h)$ extends directly to deep CNNs. Each layer in a CNN – convolution, nonlinearity, pooling – is Lipschitz continuous, so an $O(h)$ perturbation in the input propagates through the network with at most a multiplicative constant depending on the network depth. Thus, the rate of decay of gradient differences across scales remains $O(h)$ for all layers. Consequently, the theoretical guarantees of MGE and Full-Multiscale (variance reduction, telescopic-sum bounds, and convergence behavior) remain valid for deep architectures, with modified but bounded constants. □

## A.2 Why coarsening is better than cropping under a multiscale training framework

To compute the residual (Equation (9)) for cropping, we can invoke similar ideas developed in Appendix A.1. Let us crop a patch of size $s \times s$ from the image and let $\omega_h$ be the set of all pixels within the cropped patch with $|\omega_h| = m$. The gradient $\mathbf{g}_{\text{crop}}^h$ for the cropped patch can be written as,

$$\mathbf{g}_{\text{crop}}^h = \frac{1}{m} \sum_{\mathbf{u}_j^h \in \omega_h} \boldsymbol{\varphi}(\mathbf{u}_j^h). \tag{28}$$

Let us express the gradient on the finest mesh $h$ in Equation (21) in a different way as,

$$\mathbf{g}^h = \frac{1}{N_h} \sum_{\mathbf{u}_j^h \in \omega_h} \boldsymbol{\varphi}(\mathbf{u}_j^h) + \frac{1}{N_h} \sum_{\mathbf{u}_j^h \in \omega_h^c} \boldsymbol{\varphi}(\mathbf{u}_j^h), \tag{29}$$

where $\omega_h^c$ represents the set of pixels outside the cropped patch and $|\omega_h| + |\omega_h^c| = |\Omega_h| = N_h$. Hence, the residual can be expressed as,

$$r_{\text{crop}} = \|\mathbf{g}^h - \mathbf{g}_{\text{crop}}^h\| = \left\| \frac{1}{N_h} \sum_{\mathbf{u}_j^h \in \omega_h^c} \boldsymbol{\varphi}(\mathbf{u}_j^h) - \left( \frac{1}{m} - \frac{1}{N_h} \right) \sum_{\mathbf{u}_j^h \in \omega_h} \boldsymbol{\varphi}(\mathbf{u}_j^h) \right\|. \tag{30}$$

Using the triangle inequality, we have,

$$r_{\text{crop}} \leq \frac{1}{N_h} \sum_{\mathbf{u}_j^h \in \omega_h^c} \|\boldsymbol{\varphi}(\mathbf{u}_j^h)\| + \left( \frac{1}{m} - \frac{1}{N_h} \right) \sum_{\mathbf{u}_j^h \in \omega_h} \|\boldsymbol{\varphi}(\mathbf{u}_j^h)\|. \tag{31}$$

And, finally using assumption (1), we get,

$$r_{\text{crop}} \leq \left( \frac{N_h - m}{N_h} \right) M + \left( \frac{1}{m} - \frac{1}{N_h} \right) mM = 2 \left( 1 - \frac{m}{N_h} \right) M. \tag{32}$$

This implies that,

$$\boxed{r_{\text{crop}} = \|\mathbf{g}^h - \mathbf{g}_{\text{crop}}^h\| \leq 2 \left( 1 - \frac{m}{N_h} \right) M = \mathcal{O}(1)} \tag{33}$$

Hence, the upper bound of $r_{\text{crop}}$ shrinks with increasing $m$ (as the cropped patch size becomes bigger, $m/N_h \to 1$) and is independent of $h$.

Now, using the telescoping sum for **cropping**, we have $\|\mathbf{g}^{h_{j-1}} - \mathbf{g}^{h_j}\| \leq 2\left(1 - \frac{m}{N_h}\right) M$ for $j = 2, \ldots, L$. Hence,

$$\sum_{j=2}^{L} \|\mathbf{g}^{h_{j-1}} - \mathbf{g}^{h_j}\| \leq \sum_{j=2}^{L} 2\left(1 - \frac{m}{N_h}\right) M = 2\left(1 - \frac{m}{N_h}\right) M(L-1) \tag{34}$$

Therefore, for cropping-based subsampling, we have,

$$\boxed{\sum_{j=2}^{L} \|\mathbf{g}^{h_{j-1}} - \mathbf{g}^{h_j}\| = \mathcal{O}(L)} \tag{35}$$

Hence, when using **cropping-based subsampling** in MGE, the total error is independent of the resolution $h$ but grows with the number of levels $L$ in MGE.

On the other hand, for the case of **coarsening**, the telescopic sum becomes,

$$\sum_{j=2}^{L} \|\mathbf{g}^{h_{j-1}} - \mathbf{g}^{h_j}\| \leq C(h + 2h + 4h + \ldots + 2^{L-2}h) = C(2^{L-1} - 1)h. \tag{36}$$

Therefore, for coarsening-based subsampling, we have,

$$\boxed{\sum_{j=2}^{L} \|\mathbf{g}^{h_{j-1}} - \mathbf{g}^{h_j}\| = \mathcal{O}(2^L h)} \tag{37}$$

Hence, when using **coarsening-based subsampling** in MGE, the total error goes to zero as $h \to 0$. **Hence, coarsening is better than cropping as an image subsampling strategy within a multiscale training framework.**

# B Computation of #WUwithin a multiscale framework

**Definition 1** (Working Unit (WU)). *A single working unit (WU) is defined by the computation of a model (neural network) on an input image on its original (i.e., highest) resolution.*

**Remark 1.** *To measure the number of working units (#WUs) required by a neural network and its training strategy, we measure how many evaluations of the highest resolution are required. That is, evaluations at lower resolutions are weighted by the corresponding downsampling factors. In what follows, we elaborate on how #WUs are measured.*

We now show how to measure the computational complexity in terms of #WUfor the three training strategies, Single-scale, Multiscale, and Full-Multiscale. For the Single-scale strategy, all computations happen on the finest mesh (size $h$), while for Multiscale and Full-Multiscale, the computations are performed at 4 mesh resolutions $(h, 2h, 4h, 8h)$. The computation of running the network on half resolution is $1/4$ of the cost, and every coarsening step reduces the work by an additional factor of 4. From Equation (5), we have,

$$\mathbb{E}\left[\mathbf{g}^h(\boldsymbol{\theta})\right] = \mathbb{E}\left[\mathbf{g}^h(\boldsymbol{\theta}) - \mathbf{g}^{2h}(\boldsymbol{\theta})\right] + \mathbb{E}\left[\mathbf{g}^{2h}(\boldsymbol{\theta}) - \mathbf{g}^{4h}(\boldsymbol{\theta})\right] + \mathbb{E}\left[\mathbf{g}^{4h}(\boldsymbol{\theta}) - \mathbf{g}^{8h}(\boldsymbol{\theta})\right] + \mathbb{E}\left[\mathbf{g}^{8h}(\boldsymbol{\theta})\right] \tag{38}$$

With Multiscale, the number of #WUin one iteration needed to compute the $\mathbb{E}\mathbf{g}^h(\boldsymbol{\theta})$ is given by,

$$\#\text{WU}_{\text{Multiscale}} = N_0\left(1 + \frac{1}{4}\right) + N_1\left(\frac{1}{4} + \frac{1}{16}\right) + N_2\left(\frac{1}{16} + \frac{1}{64}\right) + \frac{N_3}{64} \tag{39}$$

where $N_0, N_1, N_2$ and $N_3$ represent the batch size at different scales. With $N_1 = 2N_0$, $N_2 = 4N_0$ and $N_3 = 8N_0$, #WUfor $I$ iterations become,

$$\boxed{\#\text{WU}_{\text{Multiscale}} = \frac{37 N_0 I}{16} \approx 2.31 N_0 I} \tag{40}$$

Alternatively, seeing an equivalent amount of data, doing these same computations on the finest mesh with the Single-scale training strategy, the #WUper iteration is given by, $N_0 \times 1 + N_1 \times 1 + N_2 \times 1 + N_3 \times 1$ images in one iteration (where each term is computed at the finest scale). With $N_1 = 2N_0$, $N_2 = 4N_0$, and $N_3 = 8N_0$, the total #WUfor $I$ iterations in this case, becomes,

$$\boxed{\#\text{WU}_{\text{Single-scale}} = 15 N_0 I} \tag{41}$$

**Thus, using Multiscale is roughly $6.5$ times cheaper than Single-scale training**.

The computation of #WUfor the Full-Multiscale strategy is more involved due to its cycle taking place at each level. As a result, #WUat resolutions $h, 2h, 4h$ and $8h$ can be computed as,

$$\#\text{WU}_{\text{Full-Multiscale}}(h) = I_h \times \left[N_0^h\left(1 + \frac{1}{4}\right) + N_1^h\left(\frac{1}{4} + \frac{1}{16}\right) + N_2^h\left(\frac{1}{16} + \frac{1}{64}\right) + \frac{N_3^h}{64}\right] \tag{42}$$

$$\#\text{WU}_{\text{Full-Multiscale}}(2h) = \frac{I_{2h}}{4} \times \left[N_0^{2h}\left(1 + \frac{1}{4}\right) + N_1^{2h}\left(\frac{1}{4} + \frac{1}{16}\right) + \frac{N_2^{2h}}{16}\right] \tag{43}$$

$$\#\text{WU}_{\text{Full-Multiscale}}(4h) = \frac{I_{4h}}{16} \times \left[N_0^{4h}\left(1 + \frac{1}{4}\right) + \frac{N_1^{4h}}{4}\right] \tag{44}$$

$$\#\text{WU}_{\text{Full-Multiscale}}(8h) = \frac{I_{8h}}{64} \times N_0^{8h} \tag{45}$$

$$\tag{46}$$

where $I_h, I_{2h}, I_{4h}$ and $I_{8h}$ represent the number of training iterations at each scale and $N_1^r = 2N_0^r$, $N_2^r = 4N_0^r$, and $N_3^r = 8N_0^r$ for each $r \in \{h, 2h, 4h, 8h\}$ with $N_0^{2^j h} = 2^j N_0$. For the denoising and deblurring tasks, we

chose $I$ iterations at the coarsest scale with $I = I_{8h} = 2I_{4h} = 4I_{2h} = 8I_h$. The total #WUfor Full-Multiscale for these two tasks is given by,

$$\text{\#WU}_{\text{Full-Multiscale}} = \frac{37}{16 \cdot 8} N_0 I + \frac{17}{32 \cdot 4} 2N_0 I + \frac{7}{64 \cdot 2} 4N_0 I + \frac{1}{64} 8N_0 I = \frac{115}{128} N_0 I \approx 0.90 N_0 I \qquad (47)$$

**Thus, it is roughly** $16$ **times more effective than using Single-scale training.** For both denoising and deblurring tasks, we chose $N_0 = 16$ and $I = 2000$.

Similarly, for the inpainting and super-resolution tasks, we chose $I$ iterations at the coarsest scale with $I = I_{8h} = I_{4h} = I_{2h} = I_h$. We observed that a larger number of iterations per level were required for these tasks to achieve a similar accuracy as the Single-scale training. The #WUfor Full-Multiscale for these two tasks is given by,

$$\text{\#WU}_{\text{Full-Multiscale}} = \frac{37}{16} N_0 I + \frac{17}{16} N_0 I + \frac{7}{16} N_0 I + \frac{1}{8} N_0 I = \frac{63}{16} N_0 I \approx 3.94 N_0 I \qquad (48)$$

**Thus, it is roughly** $3.8$ **times more effective than using Single-scale training.** For inpainting and super-resolution, we chose $N_0 = 16$ and $I = 2000$, and $N_0 = 8$ and $I = 250$, respectively.

## C    Additional results

### C.1    Experiments for the denoising task on the CelebA dataset

To observe the behavior of the Full-Multiscale algorithm, we performed additional experiments for the denoising task on the CelebA dataset using UNet and ResNet. The results have been presented in Table 3, showing that both multiscale training strategies achieved similar or better performance to single-scale training for all networks but with a considerably lower number of #WU.

Table 3: Comparison of different training strategies using UNet and ResNet for the **denoising** task on the CelebA dataset. Here, the Multiscale and Full-Multiscale training utilize only the coarsening strategy for image subsampling.

| Training strategy | #WU ($\downarrow$) | MSE ($\downarrow$) | |
| --- | --- | --- | --- |
| | | UNet | ResNet |
| Single-scale | 480k | 0.0663 | 0.0553 |
| Multiscale (Ours) | 74k | 0.0721 | 0.0589 |
| Full-Multiscale (Ours) | **28.7k** | 0.0484 | 0.0556 |

### C.2    Experiments with deeper networks

In this section, we present the results of our multiscale training strategies for deeper CNNs. To this end, we compare the training to ResNet18 and ResNet50, as well as UNet with 5 levels, for the image denoising task on the STL10 dataset. The results are presented in Table 4.

### C.3    Comparison of different training strategies under fixed computational budget

We assessed the performance (as measured by MSE) for both standard convolution under both Single-scale and Multiscale training strategies as a function of the computational budget, measured by #WU. Table 5 shows that for the same computational budget (#WU), Multiscale training consistently achieves lower error than Single-scale training across all tested budgets. At extremely low budgets (10–20 #WU), Multiscale attains more than 50% lower MSE due to its ability to allocate large batches to coarse, cheap levels, producing

Table 4: Comparison of different training strategies using ResNet18, ResNet50, and a deeper UNet for the **denoising** task on the STL10 dataset. Here, the Multiscale and Full-Multiscale training utilize only the coarsening strategy for image subsampling.

| Training Strategy | #WU ($\downarrow$) | MSE ($\downarrow$) | | |
|---|---|---|---|---|
| | | ResNet18 | ResNet50 | UNet (5 levels) |
| Single-scale | 480k | 0.1623 | 0.1614 | 0.1610 |
| Multiscale (Ours) | 74k | 0.1622 | 0.1611 | 0.1604 |
| Full-Multiscale (Ours) | **28.7k** | 0.1588 | 0.1598 | 0.1597 |

lower-variance gradient estimates. Even at higher budgets (60–90 #WU), Multiscale method still maintains its advantage. These results empirically confirm that Multiscale training makes significantly more efficient use of available compute and that reductions in computational cost do not imply reductions in performance when using our multiscale framework.

Table 5: Comparison of Single-scale and Multiscale training strategies under fixed computational budgets, as measured by #WU.

| Computational budget (#WU) | MSE ($\downarrow$) | |
|---|---|---|
| | Single-scale | Multiscale |
| 10 | 0.1000 | 0.0444 |
| 20 | 0.0514 | 0.0179 |
| 30 | 0.0354 | 0.0231 |
| 40 | 0.0277 | 0.0122 |
| 60 | 0.0204 | 0.0191 |
| 70 | 0.0178 | 0.0133 |
| 80 | 0.0159 | 0.0114 |
| 90 | 0.0151 | 0.0101 |

### C.4  Ablation over different number of resolution levels within Multiscale training

In this section, we experiment with the number of resolution levels used in our Multiscale and Full-Multiscale training strategies. We had conducted our experiments in the main text in Table 2 with 4 levels of resolutions $h, 2h, 4h$ and $8h$. The number of levels of resolution is a hyperparameter in our experiment which can be tuned on a held-out validation set. To illustrate this point, we show the performance of Multiscale and Full-Multiscale for the denoising task on the STL10 dataset in Table 6 for number of levels 4, 3 and 2. Upon going from 4 levels of resolution to 2 levels of resolution, the performance, in general, slightly degrades for all networks, although it leads to significant gains in computational savings due to results #WUfor lower number of levels of resolution. In fact, the number of iterations and batch size (additional hyperparameters in our experiments, as calculated in Appendix B), can also be tweaked further leading to different (better) values of #WUagainst performance on MSE.

## D  Experimental setting

For the denoising task, the experiments were conducted on STL10 and CelebA datasets using networks, UNet (Ronneberger et al., 2015) and ResNet (He et al., 2016). For deblurring and inpainting tasks, the experiments were conducted on STL10 and CelebA datasets, respectively, using the same two networks as the denoising task. For the deblurring experiments, we used a blurring kernel $K(x,y) = \frac{1}{2\pi\sigma_x\sigma_y} \exp\left(-\frac{x^2}{\sigma_x^2} - \frac{y^2}{\sigma_y^2}\right)$ with $\sigma_x = \sigma_y = 3$ to blur the input images. For the inpainting task, we introduced up to 3 (randomly chosen for each image) small rectangles with heights and widths sampled uniformly from the range $[s/12, s/6]$, where

Table 6: Comparison of the performance sensitivity to the number of resolution levels for Multiscale and Full-Multiscale training strategies using UNet and ResNet for the **denoising** task on the STL10 dataset. Here, the Multiscale and Full-Multiscale training utilize only the coarsening strategy for image subsampling.

| Training strategy | # of Levels | #WU ($\downarrow$) | MSE ($\downarrow$) | |
|---|---|---|---|---|
| | | | UNet | ResNet |
| Multiscale | 4 | 74k | 0.1975 | 0.1653 |
| Full-Multiscale | | 28.7k | 0.1567 | 0.1658 |
| Multiscale | 3 | 68k | 0.2010 | 0.1741 |
| Full-Multiscale | | 19.5k | 0.1644 | 0.1703 |
| Multiscale | 2 | 56k | 0.2641 | 0.2081 |
| Full-Multiscale | | 11k | 0.1895 | 0.1730 |

$s \times s$ is the dimension of the image. The key details of the training experimental setup for the denoising, deblurring, and inpainting tasks are summarized in Table 7. For the super-resolution task, the experiments were conducted on Urban100 dataset using networks, ESPCN (Shi et al., 2016) and ResNet (He et al., 2016). The key details of the training experimental setup for the super-resolution task are summarized in Table 8.

All our experiments were conducted on a system with NVIDIA A6000 GPU with 48GB of memory and Intel(R) Xeon(R) Gold 5317 CPU @ 3.00GHz with x86_64 processor, 48 cores, and a total available RAM of 819 GB. Upon acceptance, we will release our source code, implemented in PyTorch (Paszke et al., 2017).

Table 7: Experimental details for training for **denoising**, **deblurring** and **inpainting** tasks

| Component | Details |
|---|---|
| Dataset | STL10 (Coates et al., 2011) and CelebA (Liu et al., 2015). Images from both datasets were resized to a dimension of $64 \times 64$ |
| Network architectures | **UNet:** 3-layer network with 32, 64, 128 filters, and 1 residual block (res-block) per layer **ResNet:** 2-layer residual network 128 hidden channels |
| Number of training parameters | UNet: 2,537,187 ResNet: 597,699 |
| Training strategies | Single-scale, Multiscale and Full-Multiscale for all networks |
| Loss function | MSE loss |
| Optimizer | Adam (Kingma & Ba, 2014) |
| Learning rate | $5 \times 10^{-4}$ (with CosineAnnealing schedular) |
| Batch size strategy | Dynamic batch sizing is used, adjusting the batch size upwards during different stages of training for improved efficiency. For details, see Appendix B. |
| Multiscale levels | 4 |
| Iterations per level | Single-scale and Multiscale (all tasks): [2000, 2000, 2000, 2000], Full-Multiscale (denoising/deblurring): [2000, 1000, 500, 250] Full-Multiscale (inpainting): [2000, 2000, 2000, 2000] |
| Evaluation metrics | MSE for denoising and deblurring, SSIM for inpainting |

Table 8: Experimental details for training for **super-resolution** task

| Component | Details |
|---|---|
| Dataset | Urban100 (Huang et al., 2015), consisting of paired low-resolution and high-resolution image patches extracted for training and validation. |
| Network architectures | **ESPCN:** 5-layer super-resolution network with 64, 64, 32, 32, and 3 filters
**ResNet:** 9-layer ResNet-like model with 100 hidden channels per layer |
| Number of training parameters | ESPCN: 69,603
ResNet: 23,315 |
| Training strategies | Single-scale, Multiscale and Full-Multiscale for all networks |
| Loss function | Negative SSIM loss |
| Optimizer | Adam (Kingma & Ba, 2014) |
| Learning rate | $1 \times 10^{-3}$ (constant) |
| Batch size strategy | Dynamic batch sizing is used, adjusting the batch size upwards during different stages of training for improved efficiency. For details, see Appendix B. |
| Multiscale levels | 4 |
| Iterations per level | Single-scale, Multiscale, and Full-Multiscale: [250, 250, 250, 250] |
| Evaluation metrics | SSIM |

# E   Visualizations

In this section, we provide visualization of the outputs obtained from Single-scale, Multiscale and Full-Multiscale training strategies using different networks for various tasks such as image denoising, deblurring, inpainting, and super-resolution. Visualizations for the denoising task on the STL-10 dataset are provided in Figure 5 and on the CelebA dataset are provided in Figure 6. Visualizations for the deblurring task on the STL-10 dataset are provided in Figure 8, and for the inpainting task on the CelebA dataset are provided in Figure 9. Visualizations for the super-resolution task on the Urban100 dataset are provided in Figure 10.

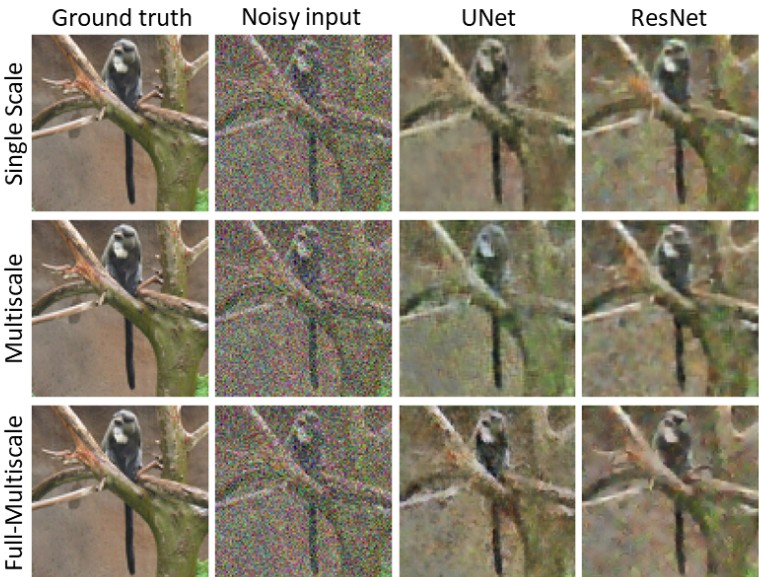

Figure 5: A comparison of different network predictions for Single-scale, Multiscale, and Full-Multiscale training for an image from the STL10 dataset for the **denoising** task. The first two columns display the original image and data (same for all rows), followed by results from UNet and ResNet. Here, the Multiscale and Full-Multiscale training utilize only the coarsening strategy for image subsampling.

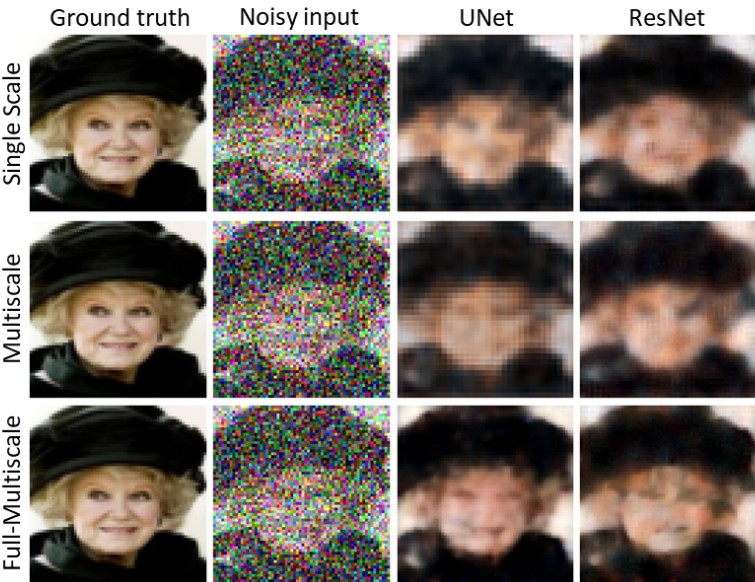

Figure 6: A comparison of different network predictions for Single-scale, Multiscale, and Full-Multiscale training for an image from the CelebA dataset for the **denoising** task. The first two columns display the original image and data (same for all rows), followed by results from UNet and ResNet. Here, the Multiscale and Full-Multiscale training utilize only the coarsening strategy for image subsampling.

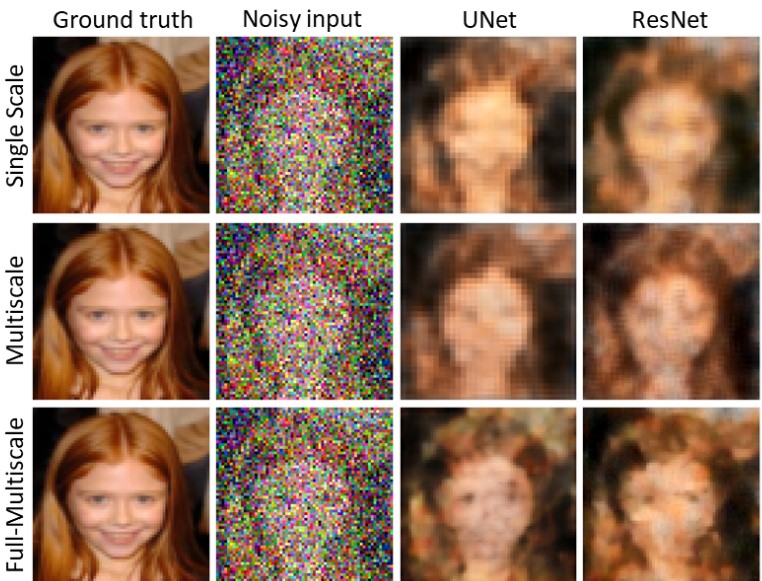

Figure 7: A comparison of different network predictions for Single-scale, Multiscale, and Full-Multiscale training for an image from the CelebA dataset for the **denoising** task. The first two columns display the original image and data (same for all rows), followed by results from UNet and ResNet. Here, the Multiscale and Full-Multiscale training utilize only the coarsening strategy for image subsampling.

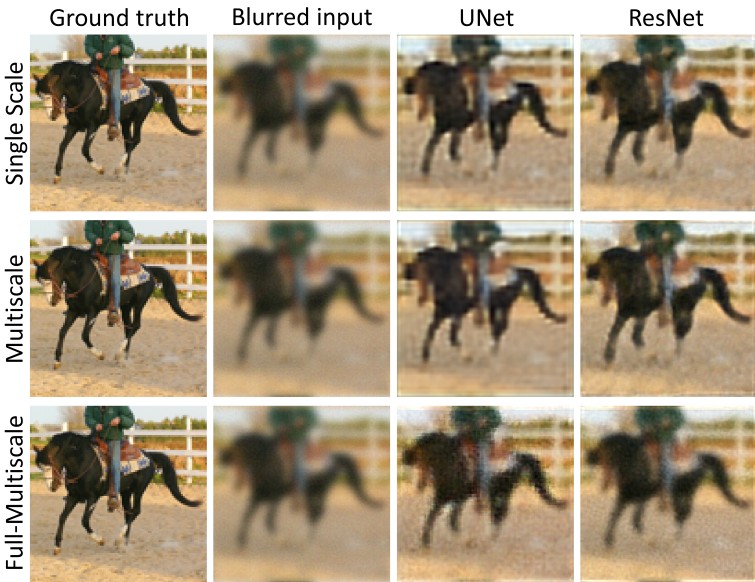

Figure 8: A comparison of different network predictions for Single-scale, Multiscale, and Full-Multiscale training for an image from the STL10 dataset for the **deblurring** task. The first two columns display the original image and data (same for all rows), followed by results from UNet and ResNet. Here, the Multiscale and Full-Multiscale training utilize only the coarsening strategy for image subsampling.

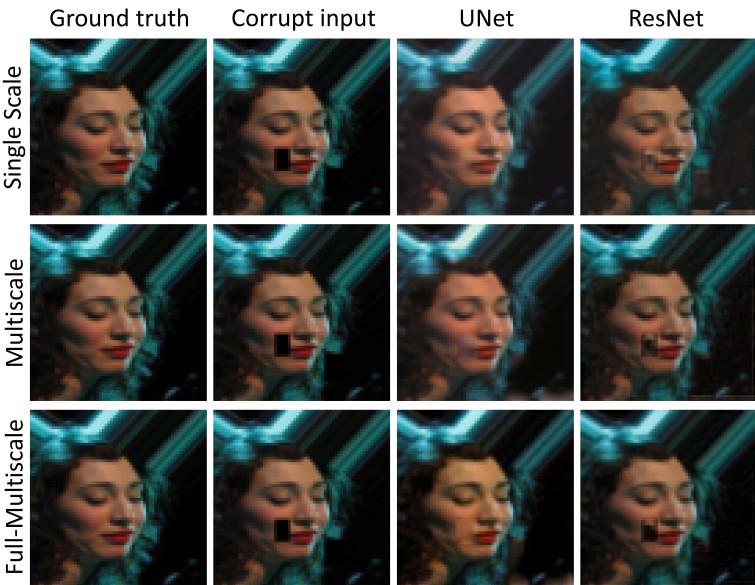

Figure 9: A comparison of different network predictions for Single-scale, Multiscale, and Full-Multiscale training for an image from the STL10 dataset for the **inpainting** task. The first two columns display the original image and data (same for all rows), followed by results from UNet and ResNet, MFC-UNet. Here, the Multiscale and Full-Multiscale training utilize only the coarsening strategy for image subsampling.

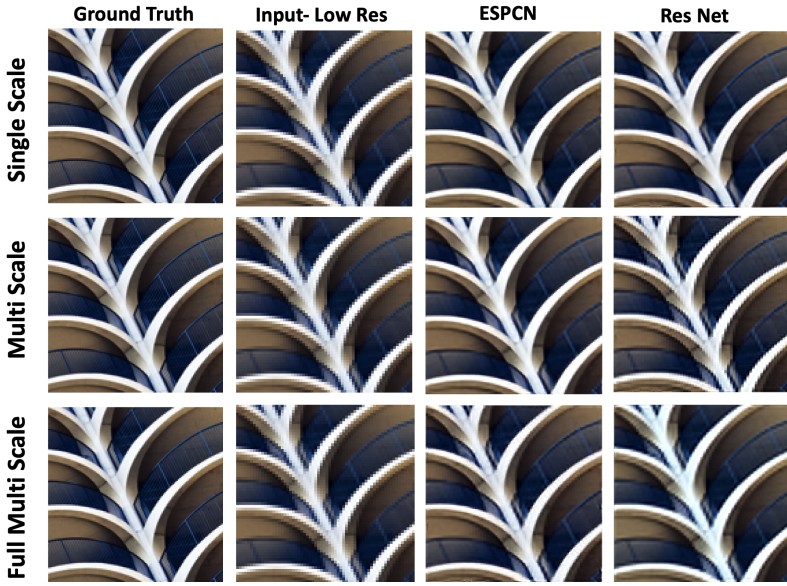

Figure 10: A comparison of different network predictions for Single-scale, Multiscale, and Full-Multiscale training for an image from the Urban100 dataset for the **super-resolution** task. The first column displays the low-resolution data (same for all rows), followed by results from ESPCN and ResNet. Here, the Multiscale and Full-Multiscale training utilize only the coarsening strategy for image subsampling.

