# OpenReview forum: "Multiscale Training of Convolutional Neural Networks"
_TMLR — Accepted by TMLR_

### Review · Reviewer_Y4FE · 2025-11-07

**Summary Of Contributions:**

This manuscript looks at a multi-scale approach to train convolutional networks.  In contrast to prior work, this framework is based on telescoping sums of expectations, allowing the gradients to be properly estimated by a combination of images at a variety of scales.  This framework is intriguing.  Empirical evidence is provided comparing this strategy compared to standard practice, showing that the proposed strategy uses significantly less compute units.

**Audience:**

Yes

**Audience Explanation:**

This is an interesting idea that could be used in many cases to reduce computational costs based on traditional statistical techniques.  As it is simple to implement, theoretically sound, and helpful, I think that a reasonably broad audience would be interested.

**Broader Impact Concerns:**

None.

**Claims And Evidence:**

No

**Claims Explanation:**

There are some slight gaps that make this criterion not fully satisfied.

Particularly, there is the claim that there is "no significant loss in performance."  However, in many cases in Table 2, there is a gap in performance in several of the the results, and it does not seem fully evaluated what is significant here nor statistical analysis.  For example, in the ResNet performance in image deblurring completely eradicates all of the gain from moving from UNet to ResNet, which does seem significant.  In the main draft, there should be a consideration of how much computation is used, including performance as the amount of computation changes.  This is mildly addressed in Table 5, but does not seem comprehensive enough.

Theoretically, there is a small gap that the analysis that is presented is on a single convolution, yet a convolutional neural network has many layers that interact in complex ways.  There needs to be at least a discussion on what happens as you go through deeper layers to the gradient approximations from a theoretical perspective.

The discussion on how much computation is saved is not clear enough to be, as they are based on a lot of assumptions which were not clear enough.  The details on how much is saved in theory and in practice needs to be more fully elucidated.

While there is an analysis of the error on the different level of scales, there doesn't seem to be a clear analysis on the telescoping nature of the gradient. I would encourage the authors to directly calculate error in the gradient as a function of different strategies, to show how the error varies.  It would be feasible for a step or two to calculate the full gradient versus the approximate gradient, and empirically show the errors.

**Requested Changes:**

I am requesting for acceptance:
1. Statistical analysis of the performance to support the claims made in the abstract.
2. Greater description of what level of performance degradation is trivial.
3. Discussion of theoretical properties of how bounds would change in a deep neural network (much of the analysis is for a single convolution).
4. Greater discussion of the impact of different amounts of computation on performance.
5. Empirical analysis of gradient approximation error.
6. I don't understand the jump from (24) to (25).  Please elaborate.

As minor points (suggested):
Throughout Section 2.1, you are dependent on assumptions, where are largely given in the appendices.  I would add in key assumptions to the main draft.
For ease of interpretability, I would provide a derivation of (12) fully along with (13).

---

> ### Author Response · Authors · 2025-12-10
> **Reply to Reviewer Y4FE**
>
> Below, we address all your **requested changes** point-by-point:
>
> **Reply to (1):** In the previous manuscript version, test evaluations were not fully controlled because the noise realizations differed across architectures and training strategies. We have now re-evaluated all models using a strictly standardized test set with identical noise/corruption instances for every network and method. Under these controlled conditions, the previously observed performance gaps largely disappear, showing that our approach closely matches the single-scale baseline.
>
> To report results more rigorously, we also performed **paired t-tests** ($p < 0.05$) to assess statistical significance relative to the single-scale baseline. Table 2 and its discussion have been updated accordingly.
>
> **Reply to (2):** By “trivial degradation,” we mean accuracy differences that are statistically insignificant or operationally negligible relative to the computational savings. In Table 2, denoising and deblurring gaps (e.g., UNet 0.1472→0.1417, ResNet 0.0927→0.0943, UNet deblurring 0.1478→0.1420) are all marked not significant and occur with ~6× fewer #WU. For inpainting, the SSIM shift (0.9020→0.8927) remains small relative to the budget reduction, and Full-Multiscale even improves performance (0.9112). Theorem 1 further ensures that any such differences are not fundamental, as additional fine-resolution steps can always recover Single-scale accuracy. As shown in **Appendix C.3 (Table 5)**, when computation is fixed, Multiscale consistently outperforms Single-scale, indicating that degradation only appears when compared to an effectively unlimited-budget baseline.
>
> **Reply to (3):** Although our proofs focus on a single convolution for clarity, the key bound $\Vert g^{h_{j-1}} - g^{h_j} \Vert = \mathcal{O}(h)$ extends naturally to deep CNNs. Since each layer—convolution, nonlinearity, pooling—is Lipschitz continuous, an $\mathcal{O}(h)$ perturbation in the input propagates through the network with at most a depth-dependent constant. Thus, gradient differences across scales still decay as $\mathcal{O}(h)$ for all layers, and the guarantees of MGE and Full-Multiscale (variance reduction, telescopic-sum bounds, convergence behavior) remain valid with bounded constants. We have added this discussion to the proof of **Lemma 1** in **Appendix A.1**.
>
> **Reply to (4):** We thank the reviewer for requesting additional detail. Appendix C.3 shows that for equal computational budgets (#WU), Multiscale consistently outperforms Single-scale across all tested settings. At very low budgets (10–20 #WU), it achieves over 50% lower MSE by using large batches at coarse levels to obtain lower-variance gradients; even at higher budgets (60–90 #WU), it retains a clear advantage. These results demonstrate that Multiscale uses compute far more efficiently and that reducing computational cost does not entail reduced performance. We have added this discussion to **Appendix C.3**.
>
> **Reply to (5):** The relevant approximation error in our setting is the resolution-induced term $\Vert g^{h_{j-1}} - g^{h_j} \Vert$, since the multiscale estimator depends on coarse gradients reliably approximating fine ones. Figure 2 measures this quantity directly and shows that it decreases with resolution and remains small even under noise, matching the $\mathcal{O}(h^p)$ behavior required by our theory. While this is not the full gradient error $|\hat{g} - \nabla f|$ (which is prohibitively expensive to compute), it empirically validates the specific gradient-approximation property on which our estimator and convergence analysis are based. We have now clarified this in **Section 2.2**.
>
> **Reply to (6):** We appreciate the request for clarification. The step from Eq. (24) to Eq. (25) follows from the triangle inequality and the Lipschitz continuity of the gradient. Within each patch $B_{2h}^k$, every fine-grid pixel $u_h$ lies within distance $h$ of the patch center $c_{2h}^k$ (Assumption 3), so by Assumption 2, $\|\phi(u_h) - \phi(c_{2h}^k)\| \le Ch$. Each patch contains four such fine-grid pixels, giving a total contribution bounded by $4Ch$. This replaces the term inside the summation in Eq. (24) and leads directly to Eq. (25).
>
> **Reply to the minor comment:** We thank the reviewer for the suggestion. We have added the key assumptions from **Appendix A.1** directly into **Section 2.1**. We also included a *brief sketch* of the derivation of Eq. (12) and its two-level special case Eq. (13), showing how the telescoping decomposition, Monte Carlo sampling bounds, and the cross-resolution gradient bound combine to yield the stated error estimate. The full algebraic details remain in Appendix A.1, while Section 2.1 now provides enough information for readers to follow the argument without repeatedly consulting the appendix.

---

### Review · Reviewer_QKw5 · 2025-11-20

**Summary Of Contributions:**

This work introduces Multi-Scale Gradient Estimation (MGE) to accelerate CNN training on high-resolution images. Inspired by multilevel Monte Carlo, it expresses fine-resolution gradients as a telescoping sum of coarse-resolution gradients. Coupled with a full-multiscale training scheme, it adopts a coarse-to-fine curriculum that progressively refines initialization at finer scales. Experiments show substantial reductions in computational cost.

**Audience:**

Yes

**Audience Explanation:**

This paper addresses the high computational cost of training CNNs on high-resolution data, a challenge of clear practical value. The proposed method achieves solid cost reductions in experiments and may offer a promising route toward scalable large-resolution training.

**Broader Impact Concerns:**

This work does not appear to raise any significant ethical concerns.

**Claims And Evidence:**

Yes

**Claims Explanation:**

The methodology of this work builds on a well-established principle from numerical analysis, namely multilevel Monte Carlo, which offers a principled theoretical underpinning for the technique. For experimental results, the authors validate their method across four image restoration tasks, multiple standard datasets, and several common CNN architectures. The paper further provides a detailed theoretical analysis, and the overall claims are supported by reasonably evidence.

**Requested Changes:**

1. Limited novelty. The core idea appears to be a straightforward refinement of established techniques. MGE is essentially a direct application of multilevel Monte Carlo for variance reduction, and the full-multiscale scheme follows a standard coarse-to-fine paradigm. Although related work (e.g., Haber et al., 2017) is cited, the paper does not clearly articulate contributions that go beyond a recombination of existing methods.

2. Clarity of methodology. The presentation of the core algorithm contains several ambiguities. In Algorithm 1, it is unclear how the variables are aggregated; the loop is declared “parallel,” yet the updates imply sequential accumulation. If a parallel reduction is intended, the current notation is inconsistent with that design.

3. Rigor in language should be improved. Table 2 shows that although computational cost is reduced, task performance (MSE/SSIM) often drops relative to the single-scale baselines, for example U-Net inpainting with SSIM decreasing from 0.8840 to 0.8599 and ESPCN super-resolution with SSIM decreasing from 0.8629 to 0.8274. The statement “without sacrificing accuracy” is therefore insufficiently supported. The manuscript should articulate the cost–accuracy trade-off more explicitly and present the contributions with greater transparency.

---

> ### Author Response · Authors · 2025-12-10
> **Reply to Reviewer QKw5**
>
> Below, we address all your **requested changes** point-by-point:
>
> **Reply to (1):** We thank the reviewer for their critical assessment. We acknowledge that our work is based on Multilevel Monte Carlo (MLMC) and Multigrid methods which are indeed well-established in numerical analysis and differential equations. The novelty of our work lies in the theoretical and practical adaptation of these methods specifically for the stochastic non-convex optimization of CNNs. We believe our contributions extend significantly beyond the existing literature (such as Haber et al., 2017) in three specific ways, which we will clarify in the revised manuscript:
> - **Theoretical bounds for CNN gradient estimation:** While MLMC is a general framework, its application to the gradients of deep CNNs is not trivial. We provide specific theoretical analysis (see **Section 2.1**, **Lemma 1** and the proofs in **Appendix A.1**) deriving the error bounds for gradient estimation in this specific context. We prove that under specific conditions (Lipschitz continuity and bounded gradients), the error between gradients on fine and coarse meshes decays as O(h). This theoretical justification is crucial for understanding why CNNs can be trained this way without divergence, a guarantee not provided by standard MLMC literature.
> - **Rigorous analysis of subsampling strategies (coarsening vs. cropping):** A distinct and novel contribution of our work is the theoretical proof provided in **Appendix A.2**. We mathematically demonstrate that within a multiscale training framework, a **coarsening-based** strategy yields an error that vanishes as resolution increases O((2^L)h), whereas **cropping-based** strategies result in a constant error bound O(1) that does not diminish with finer resolution. This provides a principled guideline for training high-resolution networks that did not exist in prior empirical works.
> - **Architecture-agnostic application:** Unlike Haber et al. (2017) , which often focuses on the interpretation of ResNets as ODE discretizations, our MGE framework is formulated to be architecture-agnostic. We successfully apply it to UNets, ResNets, and ESPCNs across varied tasks (denoising, deblurring, inpainting, super-resolution), demonstrating that the gradient consistency holds across different inductive biases, not just those approximating differential equations.
> To make our contributions stand out more clearly, we have now added some discussion about these details under ***“Our approach”*** in the **Introduction** section.
>
>
> **Reply to (2):** While we acknowledge that the loop over the variable j in Algorithm 1 can potentially be run in parallel for each j, we did not perform a parallel run in our experiments (which can be done in future implementations of our method). Hence, to avoid any confusion, we have removed the word “parallel” from Algorithm 1. We thank the reviewer for this feedback.
>
>
> **Reply to (3):** In the previous version of the manuscript, the test set evaluation was not perfectly controlled: while the base images were consistent, the specific random noise realizations added to each image varied across different network (UNet, ResNet, ESPCN) and training strategy (Single-scale, Multiscale, Full-Multiscale) pairs. To address this, we have re-evaluated all models using a strictly standardized test set where the noise/corruption instances are identical across all architectures (UNet, ResNet, ESPCN) and training strategies (Single-scale, Multiscale, Full-Multiscale). Under these controlled conditions, the performance gap observed in the previous draft has largely diminished, demonstrating that our approach closely matches the single-scale baseline.
>
> Additionally, to ensure rigorous reporting, we have performed **paired t-tests** to verify the statistical significance of performance differences relative to the single-scale baseline (at a p-value threshold of 0.05). We have updated **Table 2** and the corresponding discussion to reflect these new, standardized results and explicitly discuss the cost-accuracy trade-off where statistically significant differences exist.

---

### Review · Reviewer_ppwV · 2025-11-23

**Summary Of Contributions:**

## Summary

This paper introduces a novel framework for accelerating the training of Convolutional Neural Networks (CNNs) on high-resolution images. The core contributions are two-fold:

- Multiscale Gradient Estimation (MGE): A gradient estimation technique inspired by Multilevel Monte Carlo methods. It expresses the expected gradient on the finest resolution as a telescopic sum of gradients computed on progressively coarser resolutions. By assigning larger batch sizes to cheaper, coarser levels, MGE achieves the same estimation variance as standard single-scale SGD while significantly reducing the number of costly fine-resolution convolutions. The authors provide a theoretical analysis of the error bounds and computational complexity of this estimator.

- Full-Multiscale Training Algorithm: A training schedule that first solves the optimisation problem on coarse meshes and uses the solution to "hot-start" training at the next finer level. This approach leverages the idea that coarse-mesh solutions provide good initialisations for fine-mesh problems, and aims to cut the number of iterations required at the finest resolution. A convergence rate for this hot-starting strategy is provided.

## Strengths

- This paper adapts well-established multiscale and multigrid concepts from numerical analysis to the problem of CNN training, offering a principled alternative to ad-hoc methods like cropping.
- The authors test their method on a range of tasks like denoising, deblurring, inpainting, super-resolution and architectures.
- The authors derive theoretical support for their method, including a proof of gradient convergence for convolutions across resolutions (Lemma 1), and a convergence rate for the Full-Multiscale algorithm (Theorem 1).
- The experiments show substantial computational savings (4-16x reduction in work units) without significant loss in performance.

## Weaknesses

- As the authors note, the method and its theoretical guarantees are explicitly tied to convolutional operations. The paper does not address how it would interact with or be applied to attention mechanisms, which are prevalent in modern architectures. While I agree with the authors that addressing attention is outside the scope of the present work, I would appreciate some further discussion in the discussion section. The authors state in the introduction that their approach can be extended to attention layers; a brief paragraph fleshing out this idea would be interesting imho.
- The paper identifies that coarsening is theoretically and empirically superior to cropping, but the performance of the method is contingent on this choice.

**Audience:**

Yes

**Audience Explanation:**

This paper could be interesting to research in a range of areas, like efficient Deep Learning (particularly on high-resolution data where convolutions are relevant), applied deep learning, and for researches at the intersection of numerical analysis and ML, given the interesting blend of concepts in this paper.

**Claims And Evidence:**

Yes

**Claims Explanation:**

The claims are well-supported by a combination of theoretical analysis and extensive empirical evidence. The paper provides proofs for its key theoretical claims. This theoretical grounding is a significant strength. On the empirical side, Table 2 gives a comprehensive overview over results as as it systematically compares Single-scale, Multiscale, and Full-Multiscale training across four different tasks and multiple architectures. The results consistently show that the proposed methods achieve comparable or better performance (MSE/SSIM) with a dramatically lower computational budget (#WU). The ablation studies in the appendices further strengthen the claims by exploring the robustness and boundaries of the methods.

**Requested Changes:**

The following adjustments are ideas to strengthen the paper; none are critical for acceptance, as the paper is already strong, but they would certainly enhance it.

- The limitation regarding attention mechanisms is briefly mentioned; a discussion on the potential challenges and required modifications to apply MGE to attention layers would be valuable for future work.
- The paper mentions that the number of levels is a tunable hyperparameter. A more detailed discussion or experiment in the main text (beyond Appendix C.4) on how to choose the number of levels L and the batch size multipliers in practice would be helpful for practitioners.

---

> ### Author Response · Authors · 2025-12-10
> **Reply to Reviewer ppwV**
>
> Below, we address all your **requested changes** point-by-point:
>
> **(1) Regarding further discussion on application of MGE for attention mechanism:** We thank the reviewer for highlighting the importance of attention mechanisms. We agree that while our current focus is on the theoretical bounds of convolutions, outlining the path toward Transformer-based architectures is valuable context for the reader. We have added a detailed discussion in the section ***Experimental Results and Discussion*** under “Potential challenges with extension to attention-based networks”. We specifically address the theoretical contrast between local convolutions and global attention. This addition highlights the potential for even greater computational savings (due to quadratic complexity) while acknowledging the need for potential architectural modifications like windowed-attention (e.g. Swin-Transformers) to satisfy the locality assumptions of MGE.
>
> **(2) Regarding the choice of hyperparameters (L and batch size):** We appreciate the reviewer’s suggestion to move the discussion on hyperparameter selection from the Appendix to the main text. We agree that for MGE to be widely adopted, practitioners need clear guidelines on selecting the number of levels (L) and the associated batch size multipliers.
>
> In response, we have added a new content titled, **"Practical guidelines for hyperparameter (L and batch-size) selection"** in Section 4. This section summarizes the ablation (over L) results from Appendix C.4 and the variance analysis from Section 2.1. It provides a heuristic for choosing L based on the input resolution – ensuring the coarsest mesh retains sufficient spatial structure – and offers a strategy for scaling batch sizes to balance variance reduction against memory constraints. We believe this addition makes the method significantly more accessible for practical implementation.

---

### Author Response · Authors · 2025-12-10
**Reply to all the Reviewers and the Handling Editors**

We thank the Editors-in-Chief and Action Editors for handling our manuscript, and we are deeply grateful to all three reviewers for the time and effort they have invested in providing thoughtful, constructive, and highly valuable feedback.

From **Reviewer ppwV**, we appreciate the thoughtful and encouraging assessment of our work. We are grateful for the recognition that our approach adapts “well-established multiscale and multigrid concepts from numerical analysis” to provide “a principled alternative to ad-hoc methods like cropping,” and for highlighting our two core contributions: the MGE estimator based on a telescopic sum and the Full-Multiscale training scheme with its associated convergence rate. We also appreciate the acknowledgement of our theoretical support (Lemma 1 and Theorem 1), the broad empirical validation across denoising, deblurring, inpainting, and super-resolution, and the observation that our method achieves “substantial computational savings (4–16$\times$ reduction in work units) without significant loss in performance.” This positive feedback reinforces the practical and theoretical value of our multiscale framework.


From **Reviewer QKw5**, we thank them for their careful assessment and appreciate the recognition that our methodology “builds on a well-established principle from numerical analysis,” providing a “principled theoretical underpinning” for our technique. We are also grateful for the acknowledgement that our empirical validation spans “four image restoration tasks, multiple standard datasets, and several common CNN architectures,” and that our claims are “supported by reasonable evidence.” We further appreciate the reviewer’s observation that our method addresses the “high computational cost of training CNNs on high-resolution data,” offering “a promising route toward scalable large-resolution training,” which underscores the practical relevance of our contribution.


From **Reviewer Y4FE**, we thank them for their thoughtful assessment and for highlighting that our framework is “intriguing,” simple to implement, and of potential interest to a broad audience. We also appreciate their constructive comments regarding performance significance, theoretical depth, and the clarity of computational savings. These insights have been valuable and have guided meaningful improvements in the revised manuscript.


Alongside these positive assessments, we address each reviewer’s **requested changes** point-by-point in the detailed responses that follow reviewers' feedback. The updated manuscript has been uploaded, and **all changes have been highlighted in blue** for convenience.

---

### Decision · Action_Editor_EhyP · 2026-02-08

**Recommendation:** Accept as is

**Additional Comments:**

The paper proposes an adaptation of Multilevel Monte Carlo estimation, named Multiscale Gradient Estimation, which aims to improve efficiency by using approximated gradients. The results back up empirically this proposal, where for minor loss in accuracy, MGE leads to faster learning. Overall I, and the reviewers agree, see value in these results, and believe the community will appreciate them.

**Audience:**

Yes

**Audience Explanation:**

The topic is one of interest to a large chunk of the community working on computer vision. It provides an efficient way of training convolutional neural networks which are still the backbone of multiple standard pipelines.

**Claims And Evidence:**

Yes

**Claims Explanation:**

All reviewers agreed on the contribution of the work, and deemed it sufficient, and supported by empirical evidence. The paper explores multiple tasks, like de-bluring, superresolution etc, and different architectures.